# Seasonal and interannual dynamics of soil microbial biomass and available nitrogen in an alpine meadow in the eastern part of Qinghai-Tibet Plateau, China

Bo Xu[1,2], Jinniu Wang[1,3], Ning Wu[1,3], Yan Wu[1,3], Fusun Shi[1,3]

[1] Chengdu Institute of Biology, Chinese Academy of Sciences, Chengdu 610041, China.

[2] Aba Teachers University, Aba, Sichuan 623002, China.

[3] Key Laboratory of Mountain Ecological Restoration and Bioresource Utilization & Ecological Restoration Biodiversity Conservation Key Laboratory of Sichuan Province, Chinese Academy of Sciences, Chengdu 610041, China.

*Correspondence to*: Fusun Shi (shifs@cib.ac.cn)

**Abstract.** Soil microbial activity occurs seasonally in frozen alpine soils during cold seasons and plays a crucial role in available N pool accumulation in soil. The intra- and interannual patterns of microbial and nutrient dynamics reflect the influences of changing weather factors, and thus provide important insights into the biogeochemical cycles and ecological functions of ecosystems. We documented seasonal and interannual dynamics of soil microbial and available N in an alpine meadow in the eastern part of Qinghai-Tibet Plateau, China between April 2011 and October 2013. Soil

samples were collected in the middle of each month and were analyzed for water content, microbial biomass C (MBC) and N, dissolved organic C and N, and inorganic N; soil microbial community compositions were measured by the dilution-plate method. Fungi and actinomycetes dominated the microbial community during the non-growing seasons, and the number of bacteria increased considerably during the early growing seasons. Consistently increasing trends of MBC and available N pools were observed during the non-growing seasons. MBC sharply declined during soil thaw and

was accompanied by a peak of available N pool. Induced by soil temperatures, significant shifts in the structure and

functions of microbial communities were found during the winter-spring transition and largely contributed to microbial

reduction. Divergent seasonal dynamics of different N forms showed a complementary nutrient supply pattern during

the growing season. Similar interannual dynamics were observed between microbial biomass and available N pools, and

5   soil temperature and water condition were the primary environmental factors driving these year-to-year fluctuations.

Under the background of changing climate, the seasonal soil microbial activity and nutrient supply patterns will be

further changed, having important implications to the productivity and biodiversity of alpine ecosystems.

## 1 Copyright statement

We agree with the copyright policy of *Biogeosciences*.

## 2 Introduction

In alpine ecosystems, soil microbial activity plays a crucial role in soil C and N cycles and nutrient transformation in

frozen soils during cold seasons (Lipson et al., 1999; Murata et al., 1999; Matthew Robson et al., 2010). Unfortunately,

information on belowground microbial activity and nutrient cycles during the growing season in alpine ecosystems are

limited. Particularly, the intraannual biogeochemical cycles affected by changing seasonal weather factors in frozen

15   regions are not fully understood. The integration between the intra- and interannual patterns in soil microbial and

biogeochemical dynamics has important implications to the exploration of the current and future impacts of climate

change on the functions of alpine ecosystems (Edwards and Jefferies, 2013).

Microorganisms in alpine environments covered seasonally with snow can survive in thin unfrozen water films when

most of the soil water is frozen (Mikan et al., 2002; Edwards and Jefferies, 2013). Previous studies indicated that

substantial microbial activity exists in the alpine soils during cold seasons, even at temperatures of −5 ℃ or lower

(Brooks et al., 1996; Lipson et al., 2002; Edwards et al., 2006). Although microbial activity is limited by cold

temperatures and substrate transports (Deming, 2002; Lipson et al., 2002; Oquist et al., 2009), its cumulative effects on

organic matter decomposition in soil during long cold seasons significantly influence annual N pools in alpine

ecosystems (Lipson et al., 1999; Schmidt and Lipson, 2004; Buckeridge and Grogan, 2008). Thus, knowledge on the

microbial activity during winter can improve the understanding nutrient supplies for plants and microbes during the

subsequent growing season.

Previous studies suggested that the fungal/bacterial ratio of soil microbial community in winter is apparently higher than

that in summer (Lipson et al., 2002; Schadt et al., 2003), and significant shifts in microbial community structure and

function occur during soil thawing in alpine tundras (Lipson et al., 2002; Schadt et al., 2003; Lipson and Schmidt, 2004).

Accompanied by these changes, the rate of microbial biomass turnover increases during winter-spring transition periods

(Edwards et al., 2006; Schmidt et al., 2007; Edwards and Jefferies, 2013). Furthermore, available C substrates for the

microbial community change from winter to summer. For example, winter microbes use dead plant materials, whereas

plant root exudates supplied available C for summer microbes (Lipson et al., 2002; Schmidt et al., 2007). These microbial

community changes bewteen winter and summer might play a key role in controlling annual patterns of nutrient cycling

and plant N uptake in alpine ecoysystems (Schmidt et al., 2007; Buckeridge and Grogan, 2008).

In alpine soils, increasing microbial biomass and avilable N pools were observed during winter time, followed by a

reduction of microbial biomass during winter-spring transition when the soil thawed (Brooks et al., 1998; Lipson et al., 1999; Schmidt and Lipson, 2004; Miller et al., 2009). Moreover, the decrease of microbial biomass is linked to a pulse of N avilability when soils thaws, as observed in alpine ecosystems (Brooks et al., 1998; Lipson et al., 1999; Schmidt et al., 2007; Yang et al., 2016 ). The release of soluble N from microbial biomass during the soil thawing period provides an important avilable N source to plants, particularly in N limited ecosystems (Lipson et al., 1999; Miller et al., 2009). However, despite ample evidence of soil microbial activity and nutrient mineralization during the winter months in alpine regions (Edwards et al., 2006; Schmidt et al., 2007; Miller et al., 2009), studies on exploring the changes in microbial and N pools during the summer growing seasons in these seasonal frozen ecosystems are few (Edwards and Jefferies, 2013). Thus, the annual patterns of microbial biomass and N pools and their responses to seasnonal and interannual weather variations in alpine ecosysterms remain unclear.

In this study, we documented the seasonal dynamics of soil microbial biomass and available N for three years in an alpine meadow in the eastern part of Qinghai-Tibet Plateau of China to address the following questions: 1) What are seasonal and interannual patterns of soil microbial and available N dynamics in the alpine meadow? 2) What environmental factors affect these dynamics? 3) What are the nutrient supply patterns of different forms of available N pools in the alpine meadow soil?

## 3 Material and methods

### 3.1 Site description

The study was performed in the alpine belt of Songpan County, which belongs to the Minshan Mountain in the eastern

part of the Qinghai-Tibet Plateau, China. Records from a meteorological station (33°1′ N, 103°41′ E, 3600 m a.s.l.) near the study area showed that the average monthly air temperatures range from −7.6 ℃ in January to 15.5 ℃ in August. The annual precipitation is 718 mm, and 70 % of which occurs from June to August. The region has no absolute frost-free period, and snowfall usually occurs from late September to early May. The alpine vegetation community has rich

species composition, and dominated by different plant species at different times of the growing season. Early flowering plants, such as *Primula sikkimensis*, *Androsace umbellate*, and *Caltha palustris*, dominate the community as soon as the snow melts; *Polygonum macrophyllum*, *Ranunculus tanguticus*, and *Carex melanocephala* dominate the middle growing season; and *Saussurea hieracioides* and *Gentiana sino-ornata* usually dominate the late growing season (Xu, unpublished data, collected in 2012, 2013). The predominant soil type is Mat Cry-gelic Cambisols i.e., silty loam

inceptisol (Chinese Soil Taxonomy Research Group, 1995; Wang et al., 2016).

Study sites were located in an alpine meadow at Kaka Mountain (32°59′ N, 103°40′ E, 3980 m a.s.l.), which is a representative landscape in this region. Three sites were selected in the alpine meadow (top, middle, and bottom of the meadow), and five replicates at each site were collected. The replicates from each site were 10 m apart from each other. Given that plant roots were mainly distributed at 0–20 cm soil depth, soil sampling was only focused on this soil layer.

**3.2 Soil sampling**

Soil samples were collected on the 15th day of each month from April 2011 to October 2013. Overall, 31 sampling times were performed. Five replicates were collected at each site during each sampling time. The upper 1–2 cm of the surface materials (i.e., living plant roots and litter) were removed from the soil samples. During the winter periods, the

samples were collected by using a portable permafrost drill. The frozen soil samples were cut into little pieces ($< 1$ cm$^3$)

with a knife and hammer, and the large root and stick were removed before further determination. The soil samples

collected during the warm seasons were sieved to separate the plant materials and other fragments greater than 2 mm in

diameter. The soils were then mixed and divided into three subsamples for further analysis. All the samples were

processed at the laboratory of Chengdu Insititute of Biology, CAS, within two days of sampling.

### 3.3 Soil temperature measurement

Soil temperatures were measured at the central part of each location used for soil sampling. The soil temperature was

recorded at 10 cm depth with DS1921G Thermochron iButton data loggers (DS1921G–F5, Maxim Integrated Products,

Dallas Semiconductor Inc., Sunnyvale, CA, USA) at 1 h interval during the experimental period. The mean daily

temperature was then calculated.

### 3.4 Soil water content, microbial and nutrient analyses

One subsample was used to measure the gravimetric soil water content (SWC) after drying at 105 °C for 12 h. The soil

microbial biomass C (MBC) and N (MBN) were determined via the chloroform-fumigation extraction method (Witt et

al., 2000). Correction factors of 0.45 for C and 0.54 for N were used to convert the chloroform labile C and N to microbial

C and N (Brookes et al., 1985; Wang et al., 2016). The total colony-forming units (CFU) of bacteria, fungi, and

actinomycetes were determined via the dilution-plate method (Li, 1996; Igbinosa, 2015). A total of 10 g of measured

fresh soil subsamples were placed into a sterile jar, to which 90 mL of sterile distilled water was added, and then the jar

was covered with a sterile rubber plug and oscillated for 10 min to make a stock solution. Serial diluent was made from

the stock solution. The $10^{-5}$ and $10^{-6}$ dilution ratios of the serial diluent were selected for bacteria and actinomycetes

determination, and $10^{-2}$ and $10^{-3}$ dilution ratios for fungi determination. The selective mediums for bacteria, fungi, and

actinomycetes were beef extract peptone agar, Sabouraud dextrose agar, and Gause synthetic agar medium, respectively.

Soil diluent (1 mL) and medium (10 mL) at 45–50 ℃ were injected into the plates and cultured at 28 ℃ for 7–10 days

for bacteria and actinomycetes, and at 25 ℃ for 3–5 days for the fungi. The CFUs of different microbes were counted

under a microscope.

The total dissolved N (TDN) content was determined. Fresh soil subsamples (15 g) were measured into a beaker and

placed into a sealed vacuum dryer together with another beaker with 100 mL of chloroform. The samples were then

subjected to vacuum treatments thrice. The vacuum dryer was placed into an incubator under a temperature of 24 ℃ for

24 h and then subjected to vacuum treatment for approximately 30 min. $K_2SO_4$ (0.5 M) was added into the chloroform-

treated soil samples with a soil weight-to-extractant volume (w/v) ratio of 1 : 5 and then shaken for 1 h at 24 ℃. The

extracted solution was filtered through filter paper (0.45 μm) and stored at −20 ℃ before determination (Lu, 2000; Jones

and Willett, 2006). Then, 10 mL of the extracted solution was placed into a test tube, in which 10 mL of oxidant (NaOH-

$K_2S_2O_8$ mixed solution) was added. The resulting solution was subjected to water bath treatment at 120 ℃ for 90 min.

The TDN was then determined with an ultraviolet spectrophotometer. For the determination of available inorganic N

($NH_4^+$–N and $NO_3^-$–N), the extracted treatment solution used was similar to that used for the TDN, except that it was

not subjected to chloroform fumigation. $NH_4^+$–N and $NO_3^-$–N contents were determined via the indophenol blue

colorimetry (Sah, 1994) and ultraviolet spectrophotometry (Norman et al., 1985), respectively. Dissolved organic N

(DON) was calculated by subtracting dissolved inorganic N ($NH_4^+$–N and $NO_3^-$–N) from TDN. For the determination

of the soil dissolved organic carbon (DOC), 10 g of fresh soil subsamples were shaken with 0.5 M $K_2SO_4$ at a 1: 5 w/v

ratio for 1 h at 24 ℃, and the suspension was filtered at 0.45 μm under suction. The DOC in the extracts was then

measured through ultraviolet spectrophotometry (Lu, 2000; Jones and Willett, 2006).

**3.5 Statistical analyses**

The normal distribution and homogeneity of variance of the sample datum were analyzed with SAS 9.2 software (SAS

Institute Inc., 2008). The results met the basic requirements of variance analysis. Microbial and nutrient variables were

analyzed to test the intraannual differences between the growing season (May to October) and non-growing season

(November to April), and interannual differences among three years. Two-way ANOVA was performed with season and

year as fixed factors. Differences in the number of bacteria, fungi, and actinomycetes between the late non-growing

season (March) and early growing season (May) for two years (2012 and 2013) were determined via two-way ANOVA.

Pearson correlation analysis was then performed to analyze the correlation of the MBC of SWC with that of the DOC

during the non-growing and growing seasons. Significances were determined at the $p < 0.05$ level, and Duncan's test

was performed to analyze the significant results of the multiple comparisons.

**4 Results**

**4.1 Soil temperature and water content**

In the alpine meadow, the mean soil temperatures (at 10 cm depth) were 6.01 ℃, 7.61 ℃, and 7.06 ℃ during the three

growing seasons (May to October) from 2011 to 2013 and −1.76 °C and −2.17 °C during the two non-growing seasons

(November to April, Fig. 1). In addition, the soil was frozen (below 0 ℃) for 125 days and 165 days during 2011–2012

and 2012–2013, and the early non-growing season (November to December) of 2011–2012 had more freeze-thaw cycle

events than those of 2012–2013.

Significant seasonal and inter-annual differences in the topsoil water contents (0–20 cm depth, SWC) were observed

(Table 1). The SWC showed a decreasing trend during the growing season and increasing trend during non-growing

season (Fig. 2A), and SWC in the non-growing season was significantly higher than that in the growing season (Fig.

2B). No significant difference was observed between the SWC mean values in the non-growing season of 2011–2012

(64.73 % ±2.22 %) and those in the non-growing season of 2012–2013 (65.68 % ±4.03 %; $p > 0.05$; Fig. 2B). However,

the SWC mean values in the growing seasons on 2011–2013 were significantly different ($p < 0.05$; Fig. 2B), and the

lowest SWC was 46.43 % ±2.28 % during 2012–2013.

**4.2 Soil microbial biomass and community**

Significant differences ($p < 0.05$) between seasons and years were observed in the soils of the alpine meadow in terms

of MBC (Table 1). The annual peak of MBC occurred in the late non-growing season (March) then sharply decreased,

indicating a diminishing trend during the growing season. The MBC reached a minimum value in the late growing season

(September) then showed an increasing trend during the non-growing season (Fig. 3A). However, a significant

decreasing trend was observed in February when the soil temperatures were the lowest (below −5 °C). In addition, the

MBC values in the non-growing seasons were consistently higher than those in the growing seasons. The mean MBC

value during the non-growing season in 2012–2013 (i.e., 943.93 mg kg$^{-1}$ ±80.01 mg kg$^{-1}$) was significantly ($p < 0.05$)

higher than those in the other seasons. Meanwhile, the mean MBC value during the growing season in 2012–2013 (i.e., 143.53 mg kg$^{-1}$ ± 20.99 mg kg$^{-1}$) was the lowest (Fig. 4). The MBC during the growing season had highly significant positive correlation with the SWC ($p < 0.01$, $r = 0.62$; Table 2).

The soil MBN values had significant interannual differences ($p < 0.05$), but the seasonal difference of MBN was not

significant ($p > 0.05$; Table 1). Its seasonal and interannual dynamics were similar to those of the MBC, and its annual peak generally occurred in April or May. Furthermore, no significant difference was observed between the mean MBN values during the growing season of 2013 and those of 2011–2012 ($p > 0.05$). The lowest MBN value (72.06 mg kg$^{-1}$ ± 5.93 mg kg$^{-1}$) was observed during the growing season in 2012–2013 (Fig. 4).

Additionally, the microbial community comprised bacteria, fungi, and actinomycetes, showing a significant shift during

the winter-spring transition (March to May; $p < 0.05$; Fig. 5). The number of bacteria in May was significantly higher ($p < 0.05$) than that in March, and the number of bacteria in May 2013 (i.e., 8.25 × 10$^6$ CFU g$^{-1}$) was significantly higher ($p < 0.05$) than that in 2012 (i.e., 7.22 × 10$^6$ CFU g$^{-1}$). The numbers of fungi and actinomycetes in March were significantly higher than that in May ($p < 0.05$). The number of fungi in March 2013 (4.33 × 10$^4$ CFU g$^{-1}$) was the highest, and no significant difference was observed between the number of actinomycetes in March 2012 and that in

March 2013 ($p > 0.05$; Fig. 5).

**4.3 Soil dissolved organic carbon**

Significant interannual differences in soil DOC contents were observed, and the seasonal dynamics of DOC had no significant difference from one another ($p > 0.05$; Table 1). The DOC peaked annually occurred in May and showed a

diminishing trend during the growing season and increasing trend during the non-growing season (Fig. 6A). No

significant difference ($p > 0.05$) was observed between the DOC contents during the non-growing season in 2011–2012

(174.27 mg kg$^{-1}$ ± 32.59 mg kg$^{-1}$) and growing season in 2012–2013 (170.85 mg kg$^{-1}$ ± 41.19 mg kg$^{-1}$) but that

significantly lower than those in other seasons ($p < 0.05$; Fig. 6B). Furthermore, the DOC during the growing season had

5 highly significant positive correlation with MBC ($p < 0.01$, $r = 0.64$; Table 2).

**4.4 Soil available nitrogen**

Soil ammonium N (NH$_4^+$–N) contents showed significant seasonal and interannual differences ($p < 0.05$; Table 1). The

annual peak of the NH$_4^+$–N content occurred in the late non-growing season (April), and then sharply reduced during

the early growing season, and finally had an increasing trend during the non-growing season (Fig. 7A). The NH$_4^+$–N

content in the non-growing season was significantly higher ($p < 0.05$) than that in the growing season. The NH$_4^+$–N

content during the non-growing season in 2012–2013 (22.21 mg kg$^{-1}$ ±3.87 mg kg$^{-1}$) was significantly higher than that

in 2011–2012 (17.23 mg kg$^{-1}$ ±3.85 mg kg$^{-1}$), and no significant difference was observed among the NH$_4^+$–N contents

during the growing seasons in 2011–2013 ($p > 0.05$; Fig. 8).

Significant seasonal and interannual differences in soil nitrate N (NO$_3^-$–N) contents were observed ($p < 0.05$; Table 1).

The NO$_3^-$–N content showed an increasing trend during non-growing seasons and increased initially before decreasing

during the growing seasons (Fig. 7B). Furthermore, a significantly reducing process was observed during the soil thawing

period (April to May). The NO$_3^-$–N contents peaked annually in June while that during the non-growing season in 2011–

2012 (7.64 mg kg$^{-1}$ ±1.12 mg kg$^{-1}$) was the lowest. No significant difference was observed among the NO$_3^-$–N contents

of the other seasons ($p > 0.05$; Fig. 8).

The DON contents had significant interannual differences ($p < 0.05$), but their seasonal differences were not significant ($p > 0.05$; Table 1). In general, the peak DON content was observed in April or May, then sharply decreased during the middle and late growing season, and finally increased during the non-growing season (Fig. 7C). Furthermore, the mean DON value during the growing season in 2012–2013 (7.53 mg kg$^{-1}$ ± 1.74 mg kg$^{-1}$) was the lowest, and it was significantly lower than those in the other years ($p < 0.05$; Fig. 8).

## 5 Discussion

### 5.1 Seasonal microbial biomass and available nitrogen dynamics

Significant seasonal dynamics of the soil microbial biomass and available N pools were observed in the alpine meadow located in the eastern part of the Qinghai-Tibet Plateau for three years (Table 1; Figs. 3 and 7). Generally, the soil MBC and available N pools both increased at the beginning of the early non-growing season, and this finding is consistent with the results of the previous studies conducted in other arctic and alpine ecosystems (Lipson et al., 1999; Lipson et al., 2002; Edwards and Jefferies, 2013). This period of active microbial activity and N mineralization benefited from substrates conducive for microbial growth, particularly those supplied by the fresh plant litter inputs in autumn (Lipson et al., 1999; Nemergut et al., 2005). However, a decline of soil MBC was observed during the deeply cold period (i.e., in February when soil temperatures were below − 5 ℃). This decline implied that the temperature threshold of these cold-adapted microbial communities was at least −5 °C, and these communities retained their high activity in alpine soils during the cold periods. Thus, an accumulation of inorganic and organic N pools occurred during the long and cold non-

growing seasons in these seasonally frozen ecosystems even though the N uptakes of plants were degraded (Schimel and

Mikan, 2005; Schmidt et al., 2007; Edwards and Jefferies, 2013).

The annual peak of MBC generally occurred during the late non-growing season while the mean soil temperatures were

below 0 ℃. A modest reduction in MBC was observed in the onset of early soil thaw, and a steep decline in MBC

occurred during the late soil-thawing period while the mean soil temperatures exceeded 0 ℃. This sharp decrease in

MBC during the transition between non-growing and growing seasons was similar to the changes of MBC in other arctic

and alpine meadows during late winter and early spring (Lipson et al., 2002; Edwards et al., 2006). Previous studies

suggested several factors that contribute to the decline of MBC during the soil thawing period. First, physical changes

in soil during thawing can result in microbial cell death and release of solutes (Jefferies et al., 2010; Edwards and Jefferies,

2013). Second, depletion of soil available C and N can also lead to microbial reductions during soil thawing (Edwards

et al., 2006; Buckeridge and Grogan, 2008). Furthermore, Edwards and Jefferies (2013) hypothesized that the oxygen

availability in soils may lead to MBC reductions, because although aerobic microbial growth can still be supported in

winter, the anaerobic soil conditions are established as soils become flooded with liquid water during the late soil thaw.

However, increasing DOC and inorganic N ($NH_4^+$–N and $NO_3^-$–N) contents were observed in our study during the non-

growing season, implying that available C and N were relatively sufficient and might not restrict the microbial activity

during the winter-sping transition. This phenomenon may be closely related to the high community productivity in the

eastern part of the Qinghai-Tibet Plateau. The aboveground biomass ranges from 299.8 g m$^{-2}$ a$^{-1}$ to 475.8 g m$^{-2}$ a$^{-1}$ in

the alpine meadows on this region (Gao et.al, 2008; Yang et al., 2014) but 198 ±73.8 g m$^{-2}$ a$^{-1}$ in the paramo grassland

of Colombia (Hofstede et al., 1995) and ranges from 160 g m$^{-2}$ a$^{-1}$ to 230 g m$^{-2}$ a$^{-1}$ in the alpine meadows of the central

Rocky mountains (Walker et al., 1994; Körner, 2003). Furthermore, the soil organic matter content in the alpine meadows

of this region ranges from 69.7 g kg$^{-1}$ to 112.4 g kg$^{-1}$ (Wu and Onipchenko, 2005) but 12.8 g kg$^{-1}$ in the Alaskan tundra

and ranges from 20.3 g kg$^{-1}$ to 34.7 g kg$^{-1}$ in the alpine meadows of the Alps and Colorado (Billings and Bliss, 1959;

Körner, 2003).

Additionally, a significant difference was observed in the microbial community compositions in the non-growing seasons

and those in the growing seasons (Fig. 5). Similar to other alpine meadows, the winter microbial community was

dominated by fungi, which is more adapted to cold temperatures and utilizes complex substrates (Lipson et al., 2002;

Schadt et al., 2003). Apart from the fungi community, another important microbial community in winter soils was the

actinomycetes, which might contribute to the seasonal dynamics of the microbial biomass. Furthermore, the number of

bacteria significantly increased during the early growing season as the soils completely thawed but number of fungi and

that of actinomycetes declined considerably. This shift in the microbial community may lead to the sharp decline in MBC

during soil thaw, partly because of the C investment per unit volume in fungal cells were threefold larger than that in

bacteria cells (Buckeridge and Grogan, 2008).

In this study, the inorganic N and DON contents both showed an increasing trend during the non-growing seasnon, and

this trend was closely related to high microbial activity in the soils of this region. However, divergent dynamics among

different forms of available N were observed during the growing season (Fig. 7). A significantly increasing process of

NH$_4^+$–N was found during the early soil thaw. On the one hand, frequent and strong freeze-thaw cycles during this period

contribute to the release of unavailable $NH_4^+$–N from the organic and inorganic colloids in alpine soils (Freppaz et al.,

2007). On the other hand, the snow thawing of this period is an important source of $NH_4^+$–N (Williams and Tonnessen,

2000). At the beginning of the growing season, the $NH_4^+$–N content sharply decreased partly because of the alpine

meadow plants preferred $NH_4^+$–N (Jaeger et al., 1999; Henry and Jefferies, 2003; Gherardi et al., 2013). Moreover,

strong microbial activity in the soil requires a large amount of $NH_4^+$–N at increasing temperature (Bowman, 1992;

Schmidt and Lipson, 2004). As observed in other alpine regions (Brooks et al., 1997; Edwards et al., 2007), the $NO_3^-$–

N had a sharp decline during the soil thaw in our study, mostly because a massive amount of $NO_3^-$–N might have run

off with the snow melt water. The $NO_3^-$–N content first increased during the early growing season and then decreased

during the middle growing season as the $NH_4^+$–N content decreased. Meanwhile, the DON content slightly decreased

during the early and middle growing season and sharply decreased during the late growing season as both $NH_4^+$–N and

$NO_3^-$–N were exhausted. These results implied that although the DON may not be the main source of N pools for plants,

it is an effective supplement of the available N pool. Furthermore, the seasonal dynamics of different available N pools

showed a significant complementarity with the nutrient supply process, playing a crucial role in maintaining the rich

biodiversity of the alpine meadow ecosystem (Qin et al., 2003; Petchey and Gaston, 2006).

5.2 Interannual microbial biomass and available nitrogen dynamics

Significant year-to-year differences in microbial biomass and available N were observed across the study years. For

example, the MBC and $NH_4^+$–N contents during the non-growing season in 2012–2013 were significantly higher than

that in 2011–2012, and the MBC during the growing season in 2012–2013 was the lowest among the growing seasons

(Figs. 4 and 8). Furthermore, significant positive correlation between MBC and SWC was observed during the growing season (Table 2). This result suggested that interannual variability of soil water conditions is an important environmental driver that affects the microbial biomass in alpine meadows. First, low soil moisture in the growing season causes a decline in plant productivity (Körner, 2003), resulting in a decline of C substrates supplied by plant root exudates and

litters. Second, low soil moisture in summer leads to an increased oxidation in the surface soil, thus exerting significant influence on the microbial communities (Blodau et al., 2004), and some of these influences are retained during winter (Edwards and Jefferies, 2013). Notably, a warm and dry non-growing season was observed during 2011–2012, accompanied with frequent freeze-thaw cycles in late autumn and early winter. These environmental variations might contribute to the reduction in soil microbial biomass during the non-growing season (Larsen et al., 2002; Yanai et al.,

2004; Mellander et al., 2007; Henry, 2008). Although the extent of the influence of these environmental factors on soil microbial biomass cannot be verified, our monitoring results suggested that the soil moisture and temperature are two important environmental factors influencing the interannual dynamic of soil microbial biomass.

In the alpine meadow, organic matter decomposition and nutrient mineralization caused by soil microbial activity during the long cold season will play a crucial role in the accumulation of soil inorganic N pool (Hidy, 2003; Rinnan et al.,

2007), and the microorganism itself is also an important soil organic N pool (Lipson et al., 2002). Thus, the interannual pattern of the soil microbial biomass largely affects the year-to-year change of soil N pool. Soil $NH_4^+$–N and DON had a consistent interannual variation with soil MBC during the non-growing season. However, they showed an incompletely consistent interannual pattern during the growing season, partly because of the plant and microbe uptakes and leaching

effects. Meanwhile, for the $NO_3^-$–N, relatively small interannual variability was observed. In addition, the interannual variability of precipitation affected the interannual pattern of available inorganic N pool in the soil. The snow melt is not only an important supplement for the $NH_4^+$–N pool (Williams and Tonnessen, 2000) but also a cause of a mass of $NO_3^-$–N losses during the soil-thawing period (Brooks et al., 1997; Edwards et al., 2007). Therefore, such interannual variations

in the microbial and nutrient dynamics may become more common and pronounced in the alpine meadow in the eastern part of the Qinghai-Tibet Plateau as a result of multiple impacts of climate change, particularly increasing extreme weather events, such as winter warming and heterogeneous precipitation (Edwards and Jefferies, 2013).

## 6 Conclusions

An increasing trend of soil MBC and available N pools was found in non-growing seasons compared with growing

seasons, with a sharp decline of MBC during the soil-thawing period. Microbial activity may not be restricted by the soil available C and N in the time of soil thaw; however, a shift of microbial community induced by changing temperatures may largely contribute to this decline in MBC. Different forms of available N pools showed a divergent decreasing pattern during the growing season, suggesting that a significantly complementary pattern of nutrient supply exists among different N pools. Furthermore, the soil microorganism not only plays a crucial role in the accumulation of inorganic N

pools but also is an important soil organic N pool itself. Thus, the interannual dynamic of soil microbial biomass substantially affects the year-to-year differences in soil available N pools. According to our monitoring results, soil temperature and water condition are the primary environmental factors driving the seasonal and interannual dynamics of soil microbial biomass and available N pools. Given the changing climate of alpine ecosystems, the soil microbial

activity and nutrient supply patterns will be further changed, playing an important role in the productivity and

biodiversity of these regions. Long-term integrative studies on intra- and interannual variations of microbial and nutrient

dynamics have important implications for understanding functions of ecosystems and their responses to the

environmental change. Combined with some objective experimental studies, these research results can provide crucial

insights into the biogeochemical cycles and functions of ecosystems in the eastern part of the Qinghai-Tibet Plateau, and

their potential responses to the future climate change.

## 7 Data availability

The data set related to this study has been provided as a supplement.

## 8 Author contribution

Fusun Shi, Ning Wu, and Yan Wu designed the experiments; Bo Xu and Jinniu Wang carried field experiments out; Bo

Xu prepared the manuscript with contributions from all co-authors.

## 9 Competing interests

The authors declare that they have no conflict of interest.

## 10 Acknowledgements

The study was funded by the Key Research Projects of 13th Five Year Plan of China (2016YFC0501805).

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



**Table**

Table 1. Results from two-way ANOVA comparing growing season (May to October) and non-growing season

(November to April) values across three years of study for -SWC, -MBC, -MBN, -DOC, $NH4^+$–N, $NO_3^-$–N, and DON

5    in the alpine meadow.

| Variable | Source | df | $F$ | $p$ |
|---|---|---|---|---|
| SWC | Year | 2 | 15.68 | < 0.01 |
| | Season | 1 | 180.62 | < 0.01 |
| | Year × season | 2 | 18.29 | < 0.01 |
| MBC | Year | 2 | 48.74 | < 0.01 |
| | Season | 1 | 860.28 | < 0.01 |
| | Year × season | 2 | 61.67 | < 0.01 |
| MBN | Year | 2 | 12.35 | < 0.01 |
| | Season | 1 | 0.06 | 0.80 |
| | Year × season | 2 | 20.79 | < 0.01 |
| DOC | Year | 2 | 6.30 | 0.00 |
| | Season | 1 | 0.04 | 0.85 |
| | Year × season | 2 | 14.73 | 0.00 |
| $NH4^+$–N | Year | 2 | 7.70 | < 0.01 |
| | Season | 1 | 28.30 | < 0.01 |
| | Year × season | 2 | 0.39 | 0.53 |
| $NO_3^-$–N | Year | 2 | 3.78 | 0.03 |
| | Season | 1 | 4.34 | 0.04 |
| | Year × season | 2 | 0.18 | 0.67 |
| DON | Year | 2 | 11.67 | < 0.01 |
| | Season | 1 | 0.63 | 0.43 |
| | Year × season | 2 | 6.40 | 0.01 |





Table 2. Pearson correlations of MBC between SWC and DOC during growing and non-growing seasons

| MBC | SWC | DOC |
|---|---|---|
| Growing season | 0.62 ** | 0.64 ** |
| Non-growing season | 0.35 ** | 0.12 ns |

Note: ns, no significant difference; **, $p < 0.01$.

## Figure legends

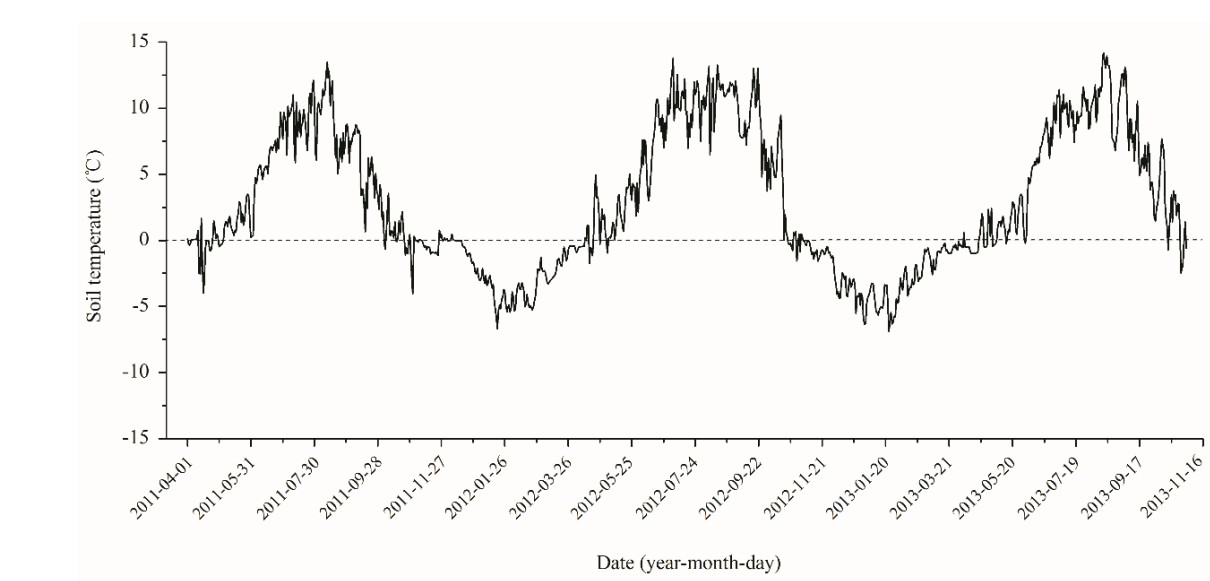

Fig. 1. Mean daily soil temperature in the alpine meadow from April 2011 to October 2013. Thermochron iButton data

loggers were placed at 10 cm soil depth to obtain automatic readings every 60 minutes, and the mean daily soil

15 temperature was calculated every day.





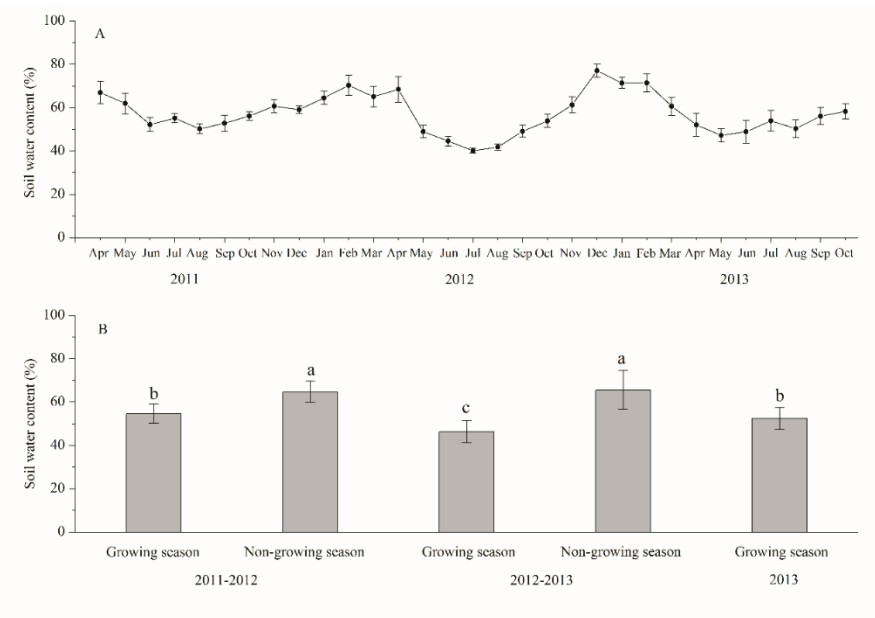

Fig. 2. Dynamics of soil water content (A; mean ±s.e.; $n = 15$) and its seasonal and interannual changes (B; mean ±s.e.;

$n = 90$) from 2011 to 2013.

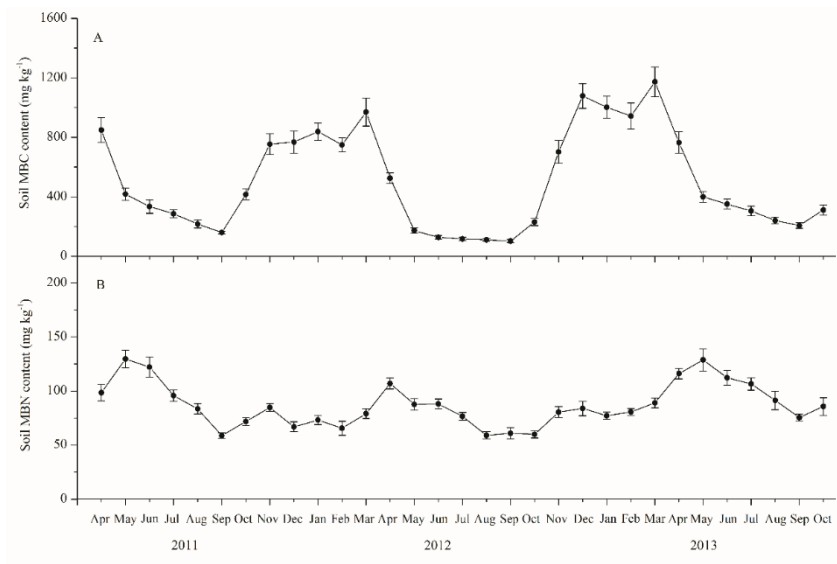

Fig. 3. Dynamics of microbial biomass C (A) and N (B) in soils of the alpine meadow from April 2011 to October 2013



(mean ±s.e.; $n = 15$).

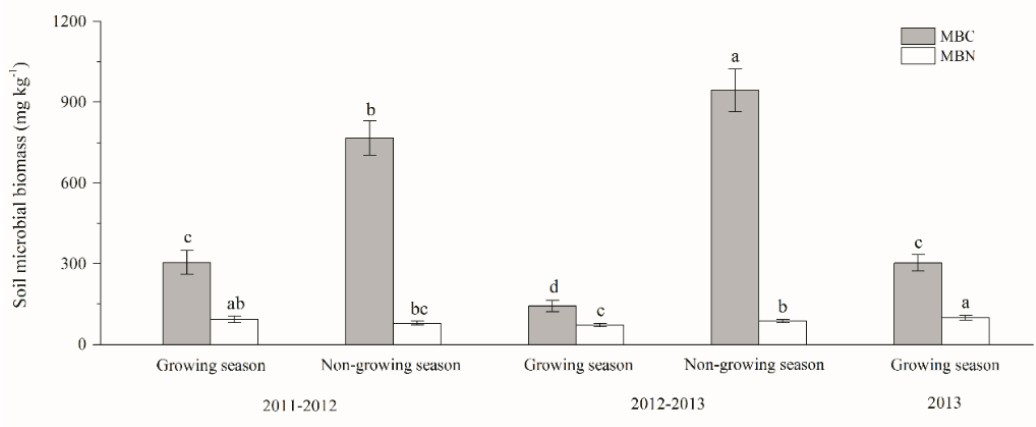

Fig. 4. Changes in microbial biomass C (MBC) and N (MBN) in the growing and non-growing seasons from 2011 to

2013 (mean ±s.e.; $n = 90$). The sampling time was on the 15th day of each month during the growing season from May

10 to October, and during the non-growing season from November to April next year. Seasons and years were compared

using two-way ANOVA, and different lowercase letters indicate significant differences ($p < 0.05$) determined via Duncan

test.





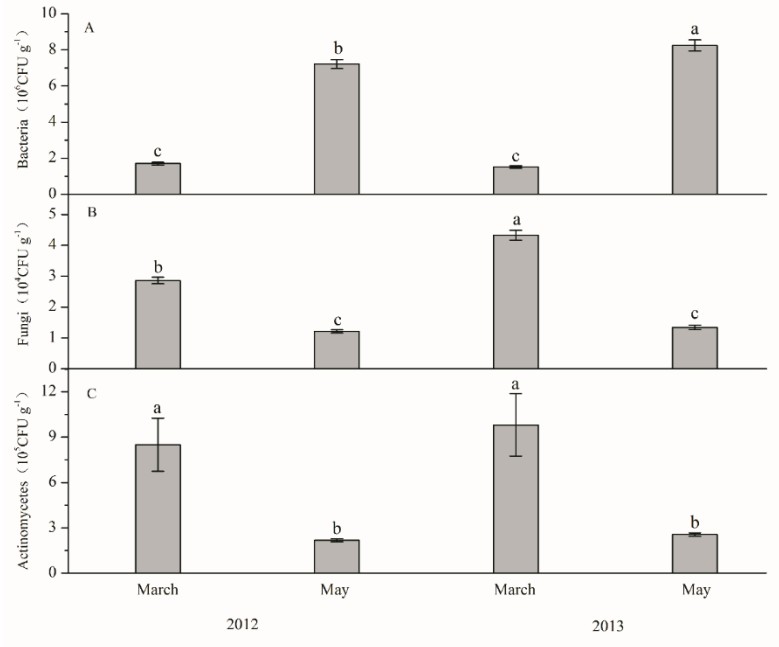

Fig. 5. Changes in the number of bacteria (A), fungi (B), and actinomycetes (C) during the transition between freezing

10    and thawing periods (mean $\pm$ s.e.; $n = 15$). The sampling time during the freezing period was on 15 March and during

the thawing period was on 15 May each year. Different lowercase letters indicate significant differences ($p < 0.05$)

according to two-way ANOVA.



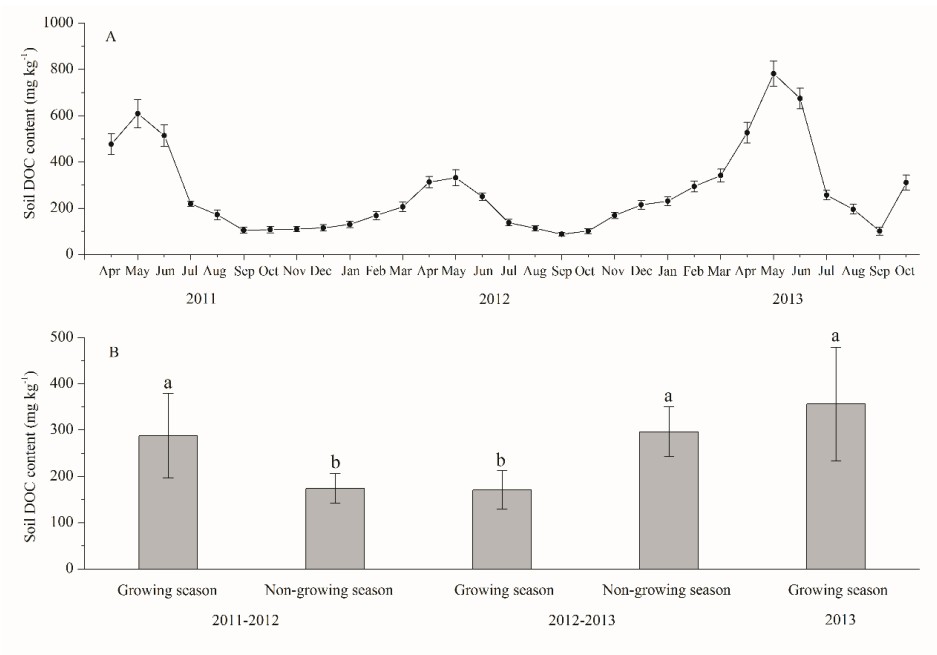

Fig. 6. Dynamics of dissolved organic C (A; mean $\pm$ s.e.; $n = 15$) and its seasonal and interannual changes (B; mean $\pm$

s.e.; $n = 90$) from 2011 to 2013.

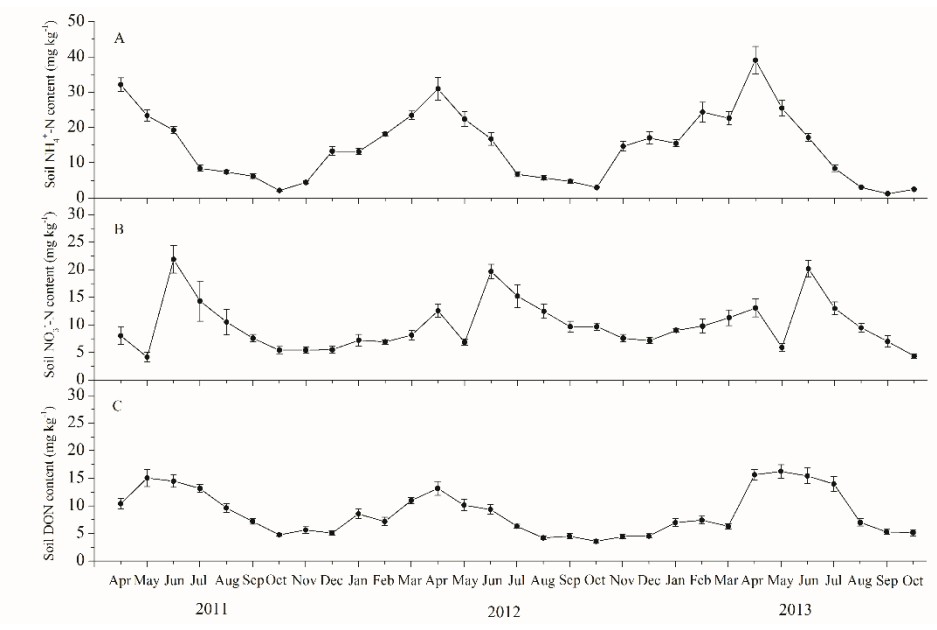





Fig. 7. Dynamics of NH$_4^+$–N(A), NO$_3^-$–N(B), and DON(C) in soils of the alpine meadow from April 2011 to October

2013 (mean ± s.e.; $n = 15$).

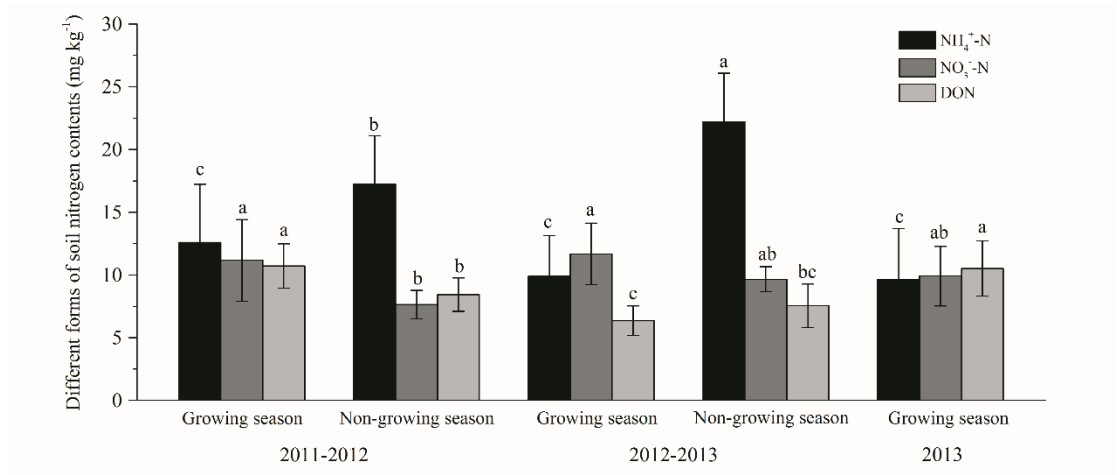

Fig. 8. Changes in NH$_4^+$–N, NO$_3^-$–N, and DON of growing and non-growing seasons from 2011 to 2013 (mean ± s.e.;

$n =90$). The sampling time was on the 15th day of each month from May to October during the growing season and

during the non-growing season from November to April next year. Seasonal and interannual differences were compared

using two-way ANOVA. Different lowercase letters indicate significant differences ($p < 0.05$) determined via Duncan

test.