# Peer review of "Seasonal and interannual dynamics of soil microbial biomass and available nitrogen in an alpine meadow in the eastern part of Qinghai–Tibet Plateau, China"

_Biogeosciences, 2017_

## Referee Comment (RC1) · Anonymous Referee #1 · 6 Apr 2017

Overall, this is a largely descriptive project, but it is well presented and the overwinter data are valuable as those types of measurements are rare. The authors might work on describing which parts of their study are most novel to help the study be better found and cited within the literature. I have some suggestions below on which topics to emphasize. The data are also remarkably "clean" for soil nutrient data with less heterogeneity of variance between dates than usual and no unusual "hot spots" of activity. The authors might discuss whether quality control measurements may have eliminated such points and if not, why the numbers are so consistent, which is not always the case for these types of studies. n = 15 is a reasonably large sample size so I do recognize that that is part of it.

[Figure]

Abstract is solid. No complaints.

INTRODUCTION I recommend the authors work to define their knowledge gaps better. There are several possible areas to discuss including location of study (including why it may or may not be different from other sites), the rarity of the overwinter measurements (there are probably just a handful of studies with this type of data), and finally, the microbial cultures are not often done in association with these types of seasonal nutrient measurements so that is worth mentioning too and describing which other studies if any have done this. The authors do mention these topics, but don't zero in on specifically what is not currently known and why it is important that we know that. I'm not saying this wasn't done at all–just that it can be done more and better.

L 15. I recommend removing these correction factors as it's widely understood that they are very ecosystem specific and hard to apply to sites in which they are not explicitly calibrated.

Three parts of the meadow were measured. Some discussion is warranted as to the spatial configuration of the sampling and why they were pooled for analysis as a single site (n = 15).

Figure 3, Fig. 7. Fig. 6B. These figures all show results that are already shown in the more detailed time courses. The authors can maybe report some of those values in the text if needed and eliminate these figures. If the authors feel this leaves the paper a little thin on figures, I would recommend exploring the relationships among the measured variables and environmental covariates using an approach such as a scatterplot matrix of correlations on a per-sample basis (ie one data point per sample, not averaged by date). Along these lines, providing the raw data as a supplement or as a link to an online repository would add value to the study.

I'm curious as to why the soil N numbers are so low-variance (particularly inorganic N). Were outliers eliminated before analysis? These types of measurements typically show substantial right skew and hot spots. Also TDN and MBN are often an order of

magnitude higher than the inorganic constituents, but that is not the case here. These points warrant discussion.

The results section is serviceable but kind of boring with its descriptions of seasonal trends and what is "significant" or not sprinkled with uninsightful p-values. I'd like to see more of a narrative structure tied to some hypotheses (eg hypothesis that there will be a crash in N availability at beginning of season as seen in other studies, a hypothesis that would be supported).

This study would benefit from a photograph of the sampled sites.

The paper is completely readable and generally well written. Still, it could use a once-over by a native speaker to fix the most challenging issues for non-native speakers such as proper preposition choice, a few cases of singluar/plural mismatch, etc.

Conclusion: keep it focused on the seasonal questions and trends. Climate change is not really addressed in any way in this study and so it's not worth mentioning here. The study's value is in its contribution to basic understanding of soil nutrient cycling seasonality.

---

## Referee Comment (RC2) · Anonymous Referee #2 · 19 Apr 2017

The paper deals with the seasonal and interannual dynamics of soil microbial biomass and available nitrogen in an alpine meadow in China. The subject is interesting but the poor english sometime let the comprehension of the text very difficult. I suggest some changes but I strongly recommend to check the english language through the assistance of a mother tongue. Moreover the paper lacks of some information such as the measurement of the snowpack depth, the estimation of the depth of the active layer and the criteria that have been used to determine the growing season lenght. Some specific points are listed below: Pag 1: lines 14/15: Did you collect topsoil samples? Please specify better line 16: add in (MBN) after and N Pag 2: line 12: With the term frozen soils do you mean permafrost soils? Pag 3: line 6: When you

mention alpine ecosystems do you mean seasonally snow cover ecosystems? Pag 4: Lines 6: again, do you refer here to subnival microbial activity during winter? Line 9: correct seasnonal into seasonal Pag 5: Line 4: When you mention frost-free periods, do you refer to air temperature? What is the mean snow depth in the area? Lines 9-10: Please add also the soil classification according to the Soil Taxonomy Lines 12-13: Do you work in a catena of soils? What do you mean with the terms top, middle and bottom? Line 14: Does this soil horizon is a A horizon? Pag 6: Line 1: In winter did you collect the soil samples under the snowpack? Lines 12-14: here you mention the chloroform fumigation technique. Why did you describe this method later at pag 7 (lines 7-15)? Pag 8: Lines 1-4: Did you fumigate also some soil samples for the determination of extractable DOC in the measurement of the microbial C? Lines 10-11: What is the definition of growing season? Did you consider the air temperature to define this period? Did you consider the soil temperature? Lines 12-13: Sorry I don't understand this sentence Pag 9: Line 2: Add respectively after 2012-2013 Lines 2-3: how do you define a freeze/thaw cycle event? Pag 10: Line 18: What do you mean from one another? Pag 11: Lines 3-4: I don't understand this sentence, in particular "but that significantly lower….." Pag 12: Line 11: Do you think is it necessary to specify "the beginning of the early non-growing season"? It's not possible to mention also the beginning of the early non-growing season? Pag 13: Lines 16: Do you mean the plant community? Please specify better this concept. Pag 14: Line 15: seasnon change into season Lines 17-18: Sorry but this sentence is not clear. What do you mean with "increasing process of NH4-N"? Pag 15: Lines 2-3: change thawing with melting. Moreover, do you have data about snow chemisty in the area? Lines 4: Preferred in comparisono to what? NO3? Line 9: During the middle growing season do you expect a high plant uptake which cause the reduction of soil inorganic N? Lines 10-11: Late in the growing season you observed a reduction in the soil inorganic N. But with the reduction of plant uptake you did not expect an opposite trend? Pag 16: Lines 7-8: Warmer and drier than 2012-2013? Moreover also a greater number of freeze/thaw cycles than 2012-2013? Is the greater number of freeze/thaw cycles recorded in the
drier season 2011-2012 related also to a thinner snowpack with a little insulation effect?

---

## Referee Comment (RC3) · Anonymous Referee #3 · 21 Apr 2017

The ms "Seasonal and interannual dynamics of soil microbial biomass and available nitrogen in an alpine meadow in the eastern part of Qinghai-Tibet Plateau, China" provides a nice dataset for microbial biomass and C and N pools at monthly intervals over 3 growing seasons and two winters in an alpine meadow. The duration of the dataset over such a long period with seasonally frozen alpine soils is quite valuable.

However, I have two important issues with this ms: 1. The justification for doing this study is not clearly formed because the research questions are not novel or clear. The background to these questions mixes Arctic references with alpine and yet is missing important references that have done very similar work in the Arctic (the Edwards 2013 paper on the long-term nutrients, which is cited, and the Buckeridge 2013 paper on

the microbes, which is not cited). The authors could fix these problems in one of two ways a) narrow their scope to alpine research and tighten their research questions, or b) include the permafrost Arctic research that they are missing that is similar to theirs and then build research questions that addresses how this research is novel within this broader framework. The best version (in my opinion) would do a bit of both options and introduce the research in both Arctic and alpine, because they are historically mixed, and then focus the paper and RQs to just alpine. The value of this study is the multi-season data in the same system. 2. The methods are unclear (why 3 sites, when are these mentioned again? Is winter vs summer sample processing associated with seasonal shift in results? Description of fumigation is confusing) and the description of the statistics is missing important details (why and how bin into seasons, and why no random factor for time?). These issues can all be fixed (I think) and a bit more effort will make this a nice paper.

Specific comments by line number, with a focus on introduction and methods since the rest may change once the introduction and methods are improved.

Introduction: P2, l17. Edwards and Jefferies is an arctic reference, not alpine P3, l1, these papers show activity, but not mechanism, and not from alpine soils (which often do not freeze deeply) - perhaps remove 'alpine' and change/add a mechanistic or review ref, such as Panikov 2006 SBB, or Jefferies 2010 SBB P3, l11, missing Buckeridge SBB 2013 here and possibly in next line (although this is not an alpine ref, but the study is very similar to this one despite focus on one year only) – then in line 13 the refs are a mix of alpine and Arctic, so it is not clear why a mix of refs would be used in some places and not others, and why this very similar study is not cited. P3, l17, again, mix of alpine and Arctic refs when alpine stated P4, l5, missing Buckeridge et al 2010 Biogeochemistry P4, l7-8, the lack of summer studies is surprising, and incorrect - there are lots of studies in the summer. Perhaps be more specific - the value of this study is a multi year investigation that encompasses both summer and winter, that is rare in alpine (Edwards and Jefferies 2013 already covered this in 2 Arctic systems).

P4, l11-14, #3 repeats #1, and how are these RQs novel? why do we need to have this information when Brooks (1998), Lipson (2002, 2004), Edwards (2006), Larsen (2007) and Buckeridge (2010, 2013) already showed this? These RQs need to be more specific about how this particular dataset advances the field. They should also be tied to the methods and results and the alpine setting – why compare seasons and years, what questions do the authors want to address by doing this?

Methods: P5, l12, I do not see these 3 sites again, just the seasonal data - where are the three sites? Were these samples pooled or only one used? P5-6, what is the snow depth and timing at these sites? P6, l1-3, the different treatment for winter (large roots removed) and summer (sieving 2mm) samples may explain different seasonal microbes and nutrient pool sizes - please indicate when this switch in handling occurred. P6, l4, are the 3 subsamples analytical replicates? P6, l8, how many iButtons for each temperature measurement? P6, l10, how was seasonal temperature calculated – by date or temperature? By date: A seasonal divide is needed – were all temp points used or were those near thaw and freeze excluded? How did the authors account for moving freeze and thaw dates across years? Or by temperature – what was the threshold, and was it based on soil or air temp? P6, l11 to p8, l4, this section is very confusing, for several reasons: the content does not match the order of the title, the TDN paragraph includes the description of fumigation, probably because the authors used the fumigation control for measuring TDN, and so they are confusing their operational process with the description. However, the biomass calculations were introduced first in the section, before the biomass extraction protocol, which is backwards. P7, l6, Does this CFU counting follow a standard protocol? Why no reference or brief protocol when so much explanation for the dilution and fumigation method? P8, l6-14, there are a few problems with this section: 1. mentioned above already, how seasonal binning of data was performed, also, it is not clear why the specific months were selected for community analysis; 2. The analyses of the independent variables (season and year) on the dependent variables should utilise a mixed-effects model with sample ID as a random effect to account for the lack of independence of samples across time.

Results: P8, l18-P9,l3, this passage describes a good reason why alpine and Arctic studies should be differentiated: these are not permafrost soils and they are not very cold. These mean 'freezing' soil temperatures are probably not experienced as freezing to a microbe full of osmolytes or a soil full of salts….although perhaps they are during extreme lows – these extreme lows should be described, in timing, depth and frequency. Are how are freeze-thaw cycles defined? What is 'more cycles' –number and dates of FT cycles should be stated for each year. P11, l16, please clarify a 'significantly reducing process' – soil redox measured with mV, or personal observation based on what criteria?

Discussion: P12, l12, again, more refs here: Brooks 1998, Edwards 2006, Larsen 2007, Buckeridge 2010 P12, l16, 'temperature threshold' for what specifically? Survival, lysis? And how does the MBC decline imply high activity in cold periods? Are the authors inferring mid-winter predation? P13, l1, 'even though' does not make sense here. P14, l9-10, the second half of this sentence is not useful P14, l18 & P16, 8, the frequency and number of freeze-thaw cycles was not stated in the results P16, l1-5, is this discussion based on gravimetric water content? Can the authors comment on why gravimetric content would correlate with non-growing season biomass if this water was frozen and unavailable? Fig.2 and associated data: are these values for gravimetric water content? How meaningful are the conclusions drawn from water pool sizes and correlations if the frozen soil water is not removed from the calculations? Fig.4 the lowercase letters represent the post-hoc test for which effect? The interaction? Fig.8, again not clear which main effect test the post-hoc letters are representing

---

## Referee Comment (RC4) · Anonymous Referee #4 · 22 Apr 2017

General comments

This paper describes intra-annual and inter-annual patterns in soil nutrient availability (inorganic and organic N) as well as microbial biomass and community structure in alpine tundra. The investigators sampled soils monthly over a 3 year period, including both the frozen and unfrozen periods. This is an impressive data set and I'm not aware of another published data set that is nearly as comprehensive. For this reason alone I encourage the authors to continue to work towards the publication of this data set. There are some aspects of both the methods and the interpretation of the results which I question and these aspects in particular require more attention by the authors before publication of this paper. See more specific comments below.
Specific comments

Referencing: Some of the references are inappropriate. Specifically, there are many citations which are used to support statements about alpine systems which were not conducted in alpine ecosystems (E.g. Page 2 line 17 and Page 4 line 8 Edwards and Jefferies, Page 3 line 6 Buckeridge and Grogan, Page 15 line 4 Henry and Jefferies). Some references are missing (Page 14 line 3: reference for Alaskan tundra is missing) and others did not examine the phenomena they are used to support (e.g. Edwards and Jefferies did not examine the survival of microorganisms surviving in thin water films (Page 3 line 1).

The methods are lacking some necessary details. The description of the 3 sites were vague: The sites are described as being at the "top middle and bottom of the meadow". Were there elevational differences between the sites? How far is the distance between them? Further, were the soils collected in the winter kept frozen into analysis? Finally, was TDN measured only after chloroform fumigation? This is how it is described, but then it would be impossible to measure MBC and MBN. It would also be good to report days below -5C rather than just below 0C: -5C is often reported as when microbial activity significantly slows.

I also question the methods used to determine changes in microbial community structure. The authors used total colony forming units of bacteria, fungi and actinomycetes using a plate dilution method. However, this only allows culturable bacteria to be counted. Further, they were all incubated at 25C regardless of season, when the winter samples likely should have been incubated at colder temperatures. Also, how were these #s compared over time? The results state which dates are significantly different from each other – were they pairwise comparisons? If the authors plan to use these methods to describe microbial community structure I would like to see citations indicating they are appropriate, as well as further description of the limitations of these methods.

[Figure]

Statistics: Because the same sites/plots were sampled repeatedly, a repeated measures ANOVA would be more appropriate than the 2-way ANOVA. Further, the description of the Pearson correlation analysis is not clear. I would like to see more of the results for this correlation described than just the r2 (Table 2). Also, throughout the results section I would like to see the actual statistics stated rather than just $p < 0.05$. Finally, is it possible to define a "peak" time for MBN or DON in the season when MBN did not vary seasonally? (Page 9 line 5).

Interpretation: Some of the interpretation of the results goes beyond what the results actually indicate. For example (Page 12 line 17) High microbial biomass does not mean there is high activity. Also see a reference to activity on page 14 line 16: this study did not contain any tests of microbial activity. Other conclusions require further elaboration. For example, The section on page 13 line 16 needs elaboration – Why would the decrease in MBC at thaw be related to the higher productivity and SOM in this site compared with others? Finally, there isn't direct support for many of the overall conclusions of the paper – this study can describe correlations, but not the types of conclusions described (e.g. soil microorganisms play a crucial role in accumulation of inorganic N pools)

Technical comments

The paper could use a thorough editing for English grammar: E.g. Community compositions should be community composition (Page 1 line 16) E.g. Change "Consistently increasing trends of MBC" to "Trends of consistently increasing MBC" E.g. Substrate transports should be substrate transport (Page 2 line 4)

---

## Author Comment (AC2) · 22 May 2017

The paper deals with the seasonal and interannual dynamics of soil microbial biomass and available nitrogen in an alpine meadow in China. The subject is interesting but the poor english sometime let the comprehension of the text very difficult. Response: We thank referee for the helpful comments. After discussing with co-authors, we thoroughly revised the manuscript and listed in supplement. Yes, the revised manuscript will be send to a professional language editing company for the language modification during the final revised period. I suggest some changes but I strongly recommend to check the english language through the assistance of a mother tongue. Moreover the paper lacks of some information such as the measurement of the snowpack depth,

the estimation of the depth of the active layer and the criteria that have been used to determine the growing season lenght. Response: We are sorry for the lacks of detail information on the snowpack during the study years, and we only had some datum on snowpack depth during the non-growing season in 2012-2013 (Page 5 lines 2-4). The definition of growing season were added to the revised manuscript, i.e., "the growing season (i.e., during early May to late October according to the plant phenology observation in the alpine meadow from 2011 to 2013)" and "The mean temperature of the growing season was calculated by the mean daily temperatures from 1th May to 31th October, and that of the non-growing season was calculated by the mean daily temperatures from 1th November to 30th April. " (Page 5 lines 15-16 and Page 7 lines 8-10). Some specific points are listed below: Pag 1: lines 14/15: Did you collect topsoil samples? Please specify better line 16: add in (MBN) after and N. Response: Yes, "Soil" was been changed to "Topsoil", and "(MBN)" was been added in the revised manuscript (Page 1 lines 15-16). Pag 2: line 12: With the term frozen soils do you mean permafrost soils? Response: No, the "frozen soils" here refers to the seasonal frozen soils. Pag 3: line 6: When you mention alpine ecosystems do you mean seasonally snow cover ecosystems? Response: Yes, "alpine ecosystems" in our study refers to the seasonally snow covered ecosystems. Pag 4: Lines 6: again, do you refer here to subnival microbial activity during winter? Line 9: correct seasnonal into seasonal. Response: Yes, "microbial activity" here refers to the subnival microbial activity during winter, and the "seasnonal" was corrected to "seasonal" in the revised manuscript. Pag 5: Line 4: When you mention frost-free periods, do you refer to air temperature? What is the mean snow depth in the area? Response: Yes, "frost-free periods" in our study refer to air temperatures. Some information on snow cover in the study area was added, i.e., "persistent snow cover usually occurs from late December to early April, and the mean snow depth is 16.58 cm in the study area (Xu, unpublished data, collected in 2012, 2013)" (Page 5 lines 12-14). Lines 9-10: Please add also the soil classification according to the Soil Taxonomy Lines 12-13: Do you work in a catena of soils? What do you mean with the terms top, middle and bottom? Line 14:

Does this soil horizon is a A horizon? Response: The soil classification of the area was added, i.e. "mountain dark brown soil" (Page 6 lines 2). Yes, serial soil samples were collected, and each sampling site was adjacent to each other at each sampling time. The terms of "top, middle and bottom" mean the locations of sampling sites, and we have revised this sentence as "Considering the soil spatial heterogeneity, three adjacent sites approximately 100 m apart (centered at 32°59′ N, 103°40′ E, 3980 m a.s.l.) were sampled, namely located at the upper, middle, and lower part of the alpine meadow" (Page 6 lines 5-7). Yes, the 0-20 cm horizon in our study is the A horizon. Pag 6: Line 1: In winter did you collect the soil samples under the snowpack? Lines 12-14: here you mention the chloroform fumigation technique. Why did you describe this method later at pag7 (lines 7-15)? Response: Yes, the alpine meadow was snow covered in deep winter, and the snow was swept before soil sample collecting. Because the chloroform fumigation treatment was also used for the determination of TDN. We rewrote this section, and the "3.4 Soil water content, microbial and nutrient analyses" section was divided into two sections, i.e., "3.4 Soil water content and nutrient analyses" and "3.5 Soil microbial biomass and community analyses" (Page 7 line 11 to page 9 line 5). Pag 8: Lines 1-4: Did you fumigate also some soil samples for the determination of extractable DOC in the measurement of the microbial C? Lines 10-11: What is the definition of growing season? Did you consider the air temperature to define this period? Did you consider the soil temperature? Response: No, the fumigate treatment did not use for the determination of extractable DOC in the measurement of the microbial C. The definition of the growing season is according to the plant phenology observation in the alpine meadow from 2011 to 2013, which indicated that the growing season is during May to October (Page 5 lines 15-16 and Page 7 lines 8-10). Lines 12-13: Sorry I don't understand this sentence. Response: This sentence (Lines 12-13) was revised as "Pearson correlation analysis was then performed to analyze the correlation of the MBC with SWC and of that with the DOC during the non-growing and growing seasons." (Page 10 lines 14-16). Pag 9: Line 2: Add respectively after 2012-2013 Lines 2-3: how do you define a freeze/thaw

cycle event? Response: Yes, "respectively" was added after 2012-2013. Actually, we did not measure the frequencies of freeze-thaw cycle events, and we inferred numbers of the freeze-thaw cycle event according to the mean soil temperature (0 °C or thereabout). It is unreasonable to define a freeze-thaw cycle event just according to soil temperature. So, this result was deleted in the revised manuscript. Pag 10: Line 18: What do you mean from one another? Response: "one another" was revised as "each other" (Page 13 line 5). Pag 11: Lines 3-4: I don't understand this sentence, in particular"but that significantly lower: : :.." Response: This sentence was revised as "The DOC contents during the non-growing season in 2011–2012 (174.27 mg kg−1 ± 32.59 mg kg−1) and growing season in 2012–2013 (170.85 mg kg−1 ± 41.19 mg kg−1) had no significant differences (p > 0.05), but those were significantly lower than that in other seasons (p < 0.05; Fig. 6B)." (Page 13 lines 7-9). Pag 12: Line 11: Do you think is it necessary to specify"the beginning of the early non-growing season"? It's not possible to mention also the beginning of the early non-growing season? Response: Yes, we thought it was necessary to specify and mention "the beginning of the early non-growing season", because MBC contents showed different dynamics during different periods of the non-growing season, i.e., MBC contents increased in early non-growing season, but decreased in deeply cold period, and then increased in the late non-growing season. Pag 13: Lines 16: Do you mean the plant community? Please specify better this concept. Response: Yes, the community productivity was mean the plant community productivity, and "community productivity" was been revised as "plant community productivity" (Page 16 lines 6-7). Pag 14: Line 15: seasnon change into season Lines 17-18: Sorry but this sentence is not clear. What do you mean with "increasing process of NH4-N"? Response: Yes, "seasnon" was been changed into "season" in the revised manuscript. Actually, the "increasing process of NH4+–N" was mean "increasing trend of NH4+–N", and we revised this sentence as "An obviously increasing trend of NH4+–N was found during the early soil thaw" (Page 17 line 8). Pag 15: Lines 2-3: change thawing with melting. Moreover, do you have data about snow chemisty in the area? Response: Yes, "thawing" was

changed into "melting". Sorry, we did not have the data on snow chemistry in the study area. Lines 4: Preferred in comparisono to what? NO3? Response: Yes, alpine plant preferred NH4+–N compared to NO3−–N and DON. Line 9: During the middle growing season do you expect a high plant uptake which cause the reduction of soil inorganic N? Response: Yes, we do agree with that a high plant uptake cause the reduction of soil inorganic N. Lines 10-11: Late in the growing season you observed a reduction in the soil inorganic N. But with the reduction of plant uptake you did not expect an opposite trend? Response: Actually, some late-flowering plants such as Gentiana sino-ornata usually dominate the late growing season, and they need to uptake relatively high available N for growing. We found that the DON was an effective supplement of the available N pool during the late growing season. Pag 16: Lines 7-8: Warmer and drier than 2012-2013? Moreover also a greater number of freeze/thaw cycles than 2012-2013? Response: Yes, the non-growing season in 2011-2012 was warmer and drier than that in 2012-2013. As we did not measure the frequencies of freeze-thaw cycle events, some similar literatures were cited in the revised manuscript. This sentence was revised as "Notably, a warmer and drier non-growing season was observed in 2011–2012 than that in 2012–2013, which might accompanied with more frequent freeze-thaw cycles during the early period of this season (Mellander et al., 2007; Henry, 2008)" (Page 18 lines 16-18). Is the greater number of freeze/thaw cycles recorded in the drier season 2011-2012 related also to a thinner snowpack with a little insulation effect? Response: Yes, we do agree with that the greater number of freeze-thaw cycles in the drier season may also related to a thinner snowpack with a little insulation effect. Unfortunately, we did not have detailed information on the snowpack during the study year.

Please also note the supplement to this comment:
http://www.biogeosciences-discuss.net/bg-2017-66/bg-2017-66-AC2-supplement.pdf

---

## Author Comment (AC4) · 22 May 2017

General comments: This paper describes intra-annual and inter-annual patterns in soil nutrient availability (inorganic and organic N) as well as microbial biomass and community structure in alpine tundra. The investigators sampled soils monthly over a 3 year period, including both the frozen and unfrozen periods. This is an impressive data set and I'm not aware of another published data set that is nearly as comprehensive. For this reason alone I encourage the authors to continue to work towards the publication of this data set. There are some aspects of both the methods and the interpretation of the results which I question and these aspects in particular require more attention by the authors before publication of this paper.

See more specific comments below. Response: We thank referee for the helpful comments. After discussing with co-authors, we thoroughly revised the manuscript and listed in supplement. Specific comments: Referencing: Some of the references are inappropriate. Specifically, there are many citations which are used to support statements about alpine systems which were not conducted in alpine ecosystems (E.g. Page 2 line 17 and Page 4 line 8 Edwards and Jefferies, Page 3 line 6 Buckeridge and Grogan, Page 15 line 4 Henry and Jefferies). Some references are missing (Page 14 line 3: reference for Alaskan tundra is missing) and others did not examine the phenomena they are used to support (e.g. Edwards and Jefferies did not examine the survival of microorganisms surviving in thin water films (Page 3 line 1). Response: Yes, we carefully revised these inappropriate references one by one in the new manuscript (Page 2 lines 12, 15, 18; Page 3 lines 3, 4, 7, 12; Page 4 lines 3, 10-13). The methods are lacking some necessary details. The description of the 3 sites were vague: The sites are described as being at the "top middle and bottom of the meadow". Were there elevational differences between the sites? How far is the distance between them? Response: Yes, the details of the 3 sites were added, i.e., "Considering the soil spatial heterogeneity, three adjacent sites approximately 100 m apart (centered at 32°59′ N, 103°40′ E, 3980 m a.s.l.) were sampled, namely located at the upper, middle, and lower part of the alpine meadow. Five replicates at each site were collected, and the replicates from each site were 10 m apart from each other. Fifteen samples collected from the three sites at each sampling time were then performed together for statistical analyses (n=15)." (Page 6 lines 5-9). Further, were the soils collected in the winter kept frozen into analysis? Response: Yes, the soil samples collected in the winter were stored at 0 °C before analysis, and all the samples were processed at the laboratory of Chengdu Insititute of Biology, CAS, within two days of sampling (Page 7 lines 1-2). Finally, was TDN measured only after chloroform fumigation? This is how it is described, but then it would be impossible to measure MBC and MBN. Response: No, different subsamples were used for the determinations of TDN, MBC and MBN. We rewrote this section, and the "3.4 Soil water content,

microbial and nutrient analyses" section was divided into two sections, i.e., "3.4 Soil water content and nutrient analyses" and "3.5 Soil microbial biomass and community analyses" (Page 7 line 11 to page 9 line 5). It would also be good to report days below -5C rather than just below 0C: -5C is often reported as when microbial activity significantly slows. Response: Yes, we added the results of the number of days below -5 °C in the revised manuscript (Page 11 lines 6-7). I also question the methods used to determine changes in microbial community structure. The authors used total colony forming units of bacteria, fungi and actinomycetes using a plate dilution method. However, this only allows culturable bacteria to be counted. Further, they were all incubated at 25C regardless of season, when the winter samples likely should have been incubated at colder temperatures. Also, how were these #s compared over time? The results state which dates are significantly different from each other – were they pairwise comparisons? If the authors plan to use these methods to describe microbial community structure I would like to see citations indicating they are appropriate, as well as further description of the limitations of these methods. Response: Actually, the dilution-plate method can be used to counting the CFU of bacteria, fungi, and actinomycetes by different selective mediums, i.e., beef extract peptone agar, Sabouraud dextrose agar, and Gause synthetic agar medium for the cultivation of bacteria, fungi, and actinomycetes, respectively (Li, 1996; Igbinosa, 2015) (Page 9 line 3). We thought if the cultivation temperature was too low, the visible microbial colony might hard to forming. So we referred to the methods of Li (1996), and measured the CFUs of bacteria, fungi, and actinomycetes. For analyses of the microbial community shifts during the transition between non-growing and growing seasons, the numbers of bacteria, fungi, and actinomycetes between the late non-growing season (i.e., in March) and early growing season (i.e., in May) for two years (2012 and 2013) were measured. These differences in the number of bacteria, fungi, and actinomycetes between season and year were then determined via two-way ANOVA, with season and year specified as fixed effects (Page 10 line 14). Statistics: Because the same sites/plots were sampled repeatedly, a repeated measures ANOVA would be more

appropriate than the 2-way ANOVA. Further, the description of the Pearson correlation analysis is not clear. I would like to see more of the results for this correlation described than just the $r^2$ (Table 2). Response: We thought the analyses of the independent variables (season and year) on the dependent variables should utilize a mixed-effects model with sample ID as a random effect to account for the lack of independence of samples across time. So, the mixed-effects model was performed for the analyses of the independent variables (season and year) on the dependent variables, and new statistical results were listed in Table 1(Page 10 lines 10-11; Page 29). Further, we revised the description of the Pearson correlation analysis as "Pearson correlation analysis was then performed to analyze the correlation of the MBC with SWC and of that with the DOC during the non-growing and growing seasons" (Page 10 line 15). In Table 2, information on r and p values was listed, we thought it was enough to describe the results of the correlation analysis. Also, throughout the results section I would like to see the actual statistics stated rather than just $p<0.05$. Finally, is it possible to define a "peak" time for MBN or DON in the season when MBN did not vary seasonally? (Page 9 line 5). Response: Yes, we added the actual statistics results in the two-way ANOVA analysis throughout the results section (Page 11 line 17; Page 12 lines 9-10; Page 13 lines 4-5; 13-14; Page 14 lines 2-3, 9-10), but the description of "$p<0.05$" was retained in the sections of the multiple comparison and Pearson correlation analysis. Finally, it is possible to define a "peak" time for MBN or DON according to their monthly values, and the MBN or DON had no significant seasonal differences just compared between growing and non-growing seasons. Interpretation: Some of the interpretation of the results goes beyond what the results actually indicate. For example (Page 12 line 17) High microbial biomass does not mean there is high activity. Response: Yes, the sentence "and these communities retained their high activity in alpine soils during the cold periods" was deleted in the revised manuscript (Page 15 lines 7-8). Also see a reference to activity on page 14 line 16: this study did not contain any tests of microbial activity. Response: Yes, "Lipson et al., 1999; Matthew Robson et al., 2010" were added (Page 17 lines 6-7). Other conclusions require further elaboration. For

example, the section on page 13 line 16 needs elaboration – Why would the decrease in MBC at thaw be related to the higher productivity and SOM in this site compared with others? Response: Actually, we did not get the conclusion that the decrease in MBC at thaw be related to the higher productivity and SOM in this site compared with others. But, we inferred that available C and N were relatively sufficient and might not restrict the microbial activity during the winter-spring transition, and this phenomenon may be closely related to the high plant community productivity and SOM in our study compared with others. Finally, there isn't direct support for many of the overall conclusions of the paper – this study can describe correlations, but not the types of conclusions described (e.g. soil microorganisms play a crucial role in accumulation of inorganic N pools) Response: Yes, we revised it as "soil microorganism not only has a close correlation with the accumulation of inorganic N pools" (Page 20 line 6) Technical comments: The paper could use a thorough editing for English grammar: E.g. Community compositions should be community composition (Page 1 line 16) E.g. Change "Consistently increasing trends of MBC" to "Trends of consistently increasing MBC" E.g. Substrate transports should be substrate transport (Page 2 line 4)? Response: Yes, we revised them one by one according to your comments (Page 1 lines 16, 18-19; Page 3 line 6), and the revised manuscript will be send to a professional language editing company for the language modification during the final revised period.

Please also note the supplement to this comment:
http://www.biogeosciences-discuss.net/bg-2017-66/bg-2017-66-AC4-supplement.pdf

---

## Author Response (AR1)

Dear Prof. Michael Weintraub,

Thank you very much for your helpful comments to our MS, and we have carefully and thoroughly revised the MS according to your comments. Meanwhile, after discussing with co-authors, the MS was thoroughly revised according to the reviews from the four anonymous referees. The detailed responses to the comments are as follows.

**Additionally, I found some of the wording to be confusing. For example, "interannual" vs. "year to year" in lines 338 and 339; and what does "incompletely consistent" mean?**

Response: Yes, we revised "year to year" as "interannual" **(Page 2 line 6; Page 18 line 18; Page 20 line 2; Page 21 line 4)**; and the "incompletely consistent" was revised as "divergent" **(Page 20 line 4).**

**Furthermore, it may be worth considering combining figures 2 and 3 into a single multipanel figure showing microbial biomass C and N temporal dynamics and growing season/non-growing season means.**

Response: Yes, figures 3 and 4 (we thought) were combined to a single multi-panel figure in the revised MS **(Page 33 Fig. 4)**.

**Also, in Figure 6, it would be nice to have every other season shaded in the top graph so it would be easier to see where each season ended.**

Response: Yes, the sections of the growing season were shaded in Fig. 7 (we thought) **(Page 36 Fig. 7)**.

**Please be sure that the clarifications provided in the response to reviewers are also incorporated into the manuscript text. Please also clearly describe how the text was revised according to the reviewer comments when submitting your revised manuscript.**

Response: Yes, the responses to reviewers were incorporated into the revised manuscript, and the detailed responses are as follows.

**Responses to Anonymous Referee #1**

**Overall, this is a largely descriptive project, but it is well presented and the overwinter data are valuable as those types of measurements are rare. The authors**

**might work on describing which parts of their study are most novel to help the study be better found and cited within the literature.**

Response: We thank referee for the helpful comments. After discussing with co-authors, we thoroughly revised the manuscript.

**I have some suggestions below on which topics to emphasize. The data are also remarkably "clean" for soil nutrient data with less heterogeneity of variance between dates than usual and no unusual "hot spots" of activity. The authors might discuss whether quality control measurements may have eliminated such points and if not, why the numbers are so consistent, which is not always the case for these types of studies. n = 15 is a reasonably large sample size so I do recognize that that is part of it.**

Response: In our study, three adjacent sites approximately 100 m apart were sampled, and five replicates at each site were collected. So fifteen soil samples were collected at each sampling time, and then the mean values of soil nutrient were calculated (n=15) (**Page 6 lines 8-13**). We thought the fifteen samples themselves would represent the heterogeneous soil nutrient status in the alpine meadow, and it might be the main reason that why you found the soil nutrient data with less heterogeneity of variance between dates. Actually, we did not take quality control to eliminate any points, and the numbers were so consistent because fifteen samples were collected at each sampling time.

**Abstract is solid. No complaints.**

Response: Thank you for your comment.

**INTRODUCTION I recommend the authors work to define their knowledge gaps better. There are several possible areas to discuss including location of study (including why it may or may not be different from other sites), the rarity of the overwinter measurements (there are probably just a handful of studies with this type of data), and finally, the microbial cultures are not often done in association with these types of seasonal nutrient measurements so that is worth mentioning too and describing which other studies if any have done this. The authors do mention these topics, but don't zero in on specifically what is not currently known and why it is important that we know that. I'm not saying this wasn't done at all–**

**just that it can be done more and better.**

Response: Yes, we revised the introduction according to your comments **(Page 3 lines 10-11; Page 4 lines 15-18)**, and we rewrote the research questions as "1) What are soil microbial and available N dynamics during the growing and non-growing seasons in the alpine meadow? 2) What are interannual patterns of soil microbial and available N dynamics in the alpine meadow? 3) What environmental factors affect these dynamics? 4) What are the relationships between soil microbial biomass and available N pools in the seasonal frozen ecosystems?" **(Page 5 lines 2-7)**

**L 15. I recommend removing these correction factors as it's widely understood that they are very ecosystem specific and hard to apply to sites in which they are not explicitly calibrated.**

Response: Thank you for your comment. But, we did not know the L15 in which page.

**Three parts of the meadow were measured. Some discussion is warranted as to the spatial configuration of the sampling and why they were pooled for analysis as a single site (n = 15).**

Response: Considering the soil spatial heterogeneity in the alpine meadow, we selected three adjacent sites for soil sampling, and five replicates at each site were collected at each sampling time. Thus, fifteen soil samples were collected at each sampling time, and then statistical analyses of soil microbial and nutrient dynamics in the alpine meadow were performed on these samples at each sampling time (n = 15) **(Page 6 lines 8-13)**.

**Figure 3, Fig. 7. Fig. 6B. These figures all show results that are already shown in the more detailed time courses. The authors can maybe report some of those values in the text if needed and eliminate these figures. If the authors feel this leaves the paper a little thin on figures, I would recommend exploring the relationships among the measured variables and environmental covariates using an approach such as a scatterplot matrix of correlations on a per-sample basis (ie one data point per sample, not averaged by date). Along these lines, providing the raw data as a supplement or as a link to an online repository would add value to the study.**

Response: We thank referee for the kindly and helpful suggestions. But we thought Fig.

3, Fig. 7, and Fig. 6B were indispensable for our study because they intuitively and detailedly showed the intra- and interannual patterns of microbial and nutrient dynamics in the alpine meadow.

**I'm curious as to why the soil N numbers are so low-variance (particularly inorganic N). Were outliers eliminated before analysis? These types of measurements typically show substantial right skew and hot spots. Also TDN and MBN are often an order of magnitude higher than the inorganic constituents, but that is not the case here. These points warrant discussion.**

Response: We did not eliminated any points before analysis. The standard error (s.e.) was used for figure drawing might be the reason why you found the soil N data with low-variance. In other ecosystems, the TDN and MBN are often an order of magnitude higher than the inorganic constituents, may because relatively high microbial activity will lead to high MBN and TDN accumulations in the soils. But in the alpine meadow ecosystems, low temperatures and N limitations may largely restrict microbial activity, causing relatively low MBN and TDN accumulations in the soils. Furthermore, alpine plants may largely uptake DON during the late growing season as the inorganic N is exhausted. We think these reasons may lead to the TDN and MBN are not an order of magnitude higher than the inorganic constituents in the alpine meadow.

**The results section is serviceable but kind of boring with its descriptions of seasonal trends and what is "significant" or not sprinkled with uninsightful p-values. I'd like to see more of a narrative structure tied to some hypotheses (eg hypothesis that there will be a crash in N availability at beginning of season as seen in other studies, a hypothesis that would be supported).**

Response: We thought you provided another paper writing habit that contain results and discussion together. But we preferred to separate the results from the discussion.

**This study would benefit from a photograph of the sampled sites.**

Response: Yes, a map of the study site was added into the revised manuscript (**Page 6 line 7 and Page 31 Fig. 1**).

**The paper is completely readable and generally well written. Still, it could use a onceover by a native speaker to fix the most challenging issues for non-native**

**speakers such as proper preposition choice, a few cases of singluar/plural mismatch, etc.**

Response: Yes, we have sent the revised manuscript to a professional language editing company for the language modification.

**Conclusion: keep it focused on the seasonal questions and trends. Climate change is not really addressed in any way in this study and so it's not worth mentioning here. The study's value is in its contribution to basic understanding of soil nutrient cycling seasonality.**

Response: In our study, we found that the interannual variations of soil temperature and water condition were the primary environmental factors driving the interannual dynamics of soil microbial biomass and available N pools. Furthermore, the alpine ecosystems are sensitive to the future climate change. So we thought it was necessary to mention the climate change in the conclusion.

**Responses to Anonymous Referee #2**

**The paper deals with the seasonal and interannual dynamics of soil microbial biomass and available nitrogen in an alpine meadow in China. The subject is interesting but the poor english sometime let the comprehension of the text very difficult. I suggest some changes but I strongly recommend to check the english language through the assistance of a mother tongue.**

Response: We thank referee for the helpful comments. After discussing with co-authors, we thoroughly revised the manuscript. Yes, the revised manuscript has been sent to a professional language editing company for the language modification during the final revised period.

**Moreover the paper lacks of some information such as the measurement of the snowpack depth, the estimation of the depth of the active layer and the criteria that have been used to determine the growing season lenght.**

Response: We are sorry for the lacks of detail information on the snowpack during the study years, and we only had some information on snowpack depth during the nongrowing season in 2012-2013 (**Page 5 lines 15-17**). The definition of growing

season were added to the revised manuscript, i.e., "the growing season (i.e., during early May to late October according to the plant phenology observation in the alpine meadow from 2011 to 2013)" **(Page 6 lines 1)** and "The mean temperature of the growing season was calculated by the mean daily temperatures from 1 May to 31 October, and that of the nongrowing season was calculated by the mean daily temperatures from 1 November to 30 April." **(Page 7 lines 11-14)**.

**Some specific points are listed below: Pag 1: lines 14/15: Did you collect topsoil samples? Please specify better line 16: add in (MBN) after and N.**

Response: Yes, "Soil" was been changed to "Topsoil", and "(MBN)" was been added in the revised manuscript **(Page 1 lines 15-16)**.

**Pag 2: line 12: With the term frozen soils do you mean permafrost soils?**

Response: No, the "frozen soils" here refers to the seasonally frozen soils.

**Pag 3: line 6: When you mention alpine ecosystems do you mean seasonally snow cover ecosystems?**

Response: Yes, "alpine ecosystems" in our study refers to the seasonally snow covered ecosystems.

**Pag 4: Lines 6: again, do you refer here to subnival microbial activity during winter? Line 9: correct seasnonal into seasonal.**

Response: Yes, "microbial activity" here refers to the subnival microbial activity during winter, and the "seasnonal" was corrected to "seasonal" in the revised manuscript.

**Pag 5: Line 4: When you mention frost-free periods, do you refer to air temperature? What is the mean snow depth in the area?**

Response: Yes, "frost-free periods" in our study refer to air temperatures. Some information on snow cover in the study area was added, i.e., "Persistent snow cover usually occurs from late December to early April, and the mean snow depth is 16.58 cm in the study area (Xu, unpublished data, collected in 2012, 2013)" **(Page 5 lines 15-17)**.

**Lines 9-10: Please add also the soil classification according to the Soil Taxonomy Lines 12-13: Do you work in a catena of soils? What do you mean with the terms top, middle and bottom? Line 14: Does this soil horizon is a A horizon?**

Response: The soil classification of the area was added, i.e. "mountain dark brown soil" **(Page 6 lines 5)**. Yes, serial soil samples were collected, and each sampling site was adjacent to each other at each sampling time. The terms of "top, middle and bottom" mean the locations of sampling sites, and we have revised this sentence as "Considering the soil spatial heterogeneity, three adjacent sites, approximately 100 m apart (centered at 32°59′ N, 103°40′ E, 3980 m a.s.l.) were selected. One site is located at the upper part of the alpine meadow, one at the middle part, and one at the lower part." **(Page 6 lines 8-10)**. Yes, the 0-20 cm horizon in our study is the A horizon.

**Pag 6: Line 1: In winter did you collect the soil samples under the snowpack? Lines 12-14: here you mention the chloroform fumigation technique. Why did you describe this method later at pag7 (lines 7-15)?**

Response: Yes, the alpine meadow was snow covered in deep winter, and the snow was swept before soil sample collecting. Because the chloroform fumigation treatment was also used for the determination of TDN. We rewrote this section, and the "3.4 Soil water content, microbial and nutrient analyses" section was divided into two sections, i.e., "3.4 Soil water content and nutrient analyses" and "3.5 Soil microbial biomass and community analyses" **(Page 7 line 15 to Page 9 line 11)**.

**Pag 8: Lines 1-4: Did you fumigate also some soil samples for the determination of extractable DOC in the measurement of the microbial C? Lines 10-11: What is the definition of growing season? Did you consider the air temperature to define this period? Did you consider the soil temperature?**

Response: No, the fumigate treatment did not use for the determination of extractable DOC in the measurement of the microbial C. The definition of the growing season is according to the plant phenology observation in the alpine meadow from 2011 to 2013, which indicated that the growing season is during May to October **(Page 6 line 1)**.

**Lines 12-13: Sorry I don't understand this sentence.**

Response: This sentence (Lines 12-13) was revised as "Pearson correlation analysis was then performed to analyze the correlation between MBC and SWC and that between MBC and DOC during the nongrowing and growing seasons." **(Page 11 lines 2-4)**.

**Pag 9: Line 2: Add respectively after 2012-2013 Lines 2-3: how do you define a freeze/thaw cycle event?**

Response: Yes, "respectively" was added after 2012-2013. Actually, we did not measure the frequencies of freeze-thaw cycle events, and we inferred numbers of the freeze-thaw cycle event according to the mean soil temperature (0 °C or thereabout). It is unreasonable to define a freeze-thaw cycle event just according to soil temperature. So, this result was deleted in the revised manuscript.

**Pag 10: Line 18: What do you mean from one another?**

Response: "one another" was revised as "each other" (**Page 13 line 5**).

**Pag 11: Lines 3-4: I don't understand this sentence, in particular〝but that significantly lower: : :.."**

Response: This sentence was revised as "The DOC contents during the nongrowing season in 2011–2012 (174.27 mg kg$^{-1}$ $\pm$ 32.59 mg kg$^{-1}$) and growing season in 2012–2013 (170.85 mg kg$^{-1}$ $\pm$ 41.19 mg kg$^{-1}$) had no significant differences ($p > 0.05$), but those were significantly lower than that in other seasons ($p < 0.05$; Fig. 6B)" (**Page 13 lines 15-18**).

**Pag 12: Line 11: Do you think is it necessary to specify〝the beginning of the early non-growing season"? It's not possible to mention also the beginning of the early non-growing season?**

Response: Yes, we thought it was necessary to specify and mention "the beginning of the early nongrowing season", because MBC contents showed different dynamics during different periods of the nongrowing season, i.e., MBC contents increased in early nongrowing season, but decreased in deeply cold period, and then increased in the late nongrowing season.

**Pag 13: Lines 16: Do you mean the plant community? Please specify better this concept.**

Response: Yes, the community productivity was mean the plant community productivity, and "community productivity" was been revised as "plant community productivity" (**Page 16 line 16**).

**Pag 14: Line 15: seasnon change into season Lines 17-18: Sorry but this sentence**

**is not clear. What do you mean with "increasing process of NH4-N"?**

Response: Yes, "seasnon" was been changed into "season" in the revised manuscript. Actually, the "increasing process of $NH_4^+$–N" was mean "increasing trend of $NH_4^+$–N", and we revised this sentence as "An obviously trend of increasing $NH_4^+$–N content was found during the early soil thaw." (**Page 17 line 18 to Page 18 line 1**).

**Pag 15: Lines 2-3: change thawing with melting. Moreover, do you have data about snow chemistry in the area?**

Response: Yes, "thawing" was changed into "melting". Sorry, we did not have the data on snow chemistry in the study area.

**Lines 4: Preferred in comparison to what? NO3?**

Response: Yes, alpine plant preferred $NH_4^+$–N compared to $NO_3^-$–N and DON.

**Line 9: During the middle growing season do you expect a high plant uptake which cause the reduction of soil inorganic N?**

Response: Yes, we do agree with that a high plant uptake causes the reduction of soil inorganic N.

**Lines 10-11: Late in the growing season you observed a reduction in the soil inorganic N. But with the reduction of plant uptake you did not expect an opposite trend?**

Response: Actually, some late-flowering plants such as *Gentiana sino-ornata* usually dominate the late growing season, and they need to uptake relatively high available N for growing. We found that the DON was an effective supplement of the available N pool during the late growing season.

**Pag 16: Lines 7-8: Warmer and drier than 2012-2013? Moreover also a greater number of freeze/thaw cycles than 2012-2013?**

Response: Yes, the nongrowing season in 2011-2012 was warmer and drier than that in 2012-2013. As we did not measure the frequencies of freeze–thaw cycle events, some similar literatures were cited in the revised manuscript. This sentence was revised as "Notably, the nongrowing season in 2011–2012 was warmer and drier than that in 2012–2013, which might accompanied with more frequent freeze–thaw cycles during the early period of this season (Mellander et al., 2007; Henry, 2008)." (**Page 19 lines**

**9-11**).

**Is the greater number of freeze/thaw cycles recorded in the drier season 2011-2012 related also to a thinner snowpack with a little insulation effect?**

Response: Yes, we do agree with that the greater number of freeze–thaw cycles in the drier season may also related to a thinner snowpack with a little insulation effect. Unfortunately, we did not have detailed information on the snowpack during the study year.

**Responses to Anonymous Referee #3**

**The ms "Seasonal and interannual dynamics of soil microbial biomass and available nitrogen in an alpine meadow in the eastern part of Qinghai-Tibet Plateau, China" provides a nice dataset for microbial biomass and C and N pools at monthly intervals over 3 growing seasons and two winters in an alpine meadow. The duration of the dataset over such a long period with seasonally frozen alpine soils is quite valuable.**

Response: We thank referee for the helpful comments. After discussing with co-authors, we thoroughly revised the manuscript.

**However, I have two important issues with this ms: 1. The justification for doing this study is not clearly formed because the research questions are not novel or clear. The background to these questions mixes Arctic references with alpine and yet is missing important references that have done very similar work in the Arctic (the Edwards 2013 paper on the long-term nutrients, which is cited, and the Buckeridge 2013 paper on the microbes, which is not cited). The authors could fix these problems in one of two ways a) narrow their scope to alpine research and tighten their research questions, or b) include the permafrost Arctic research that they are missing that is similar to theirs and then build research questions that addresses how this research is novel within this broader framework. The best version (in my opinion) would do a bit of both options and introduce the research in both Arctic and alpine, because they are historically mixed, and then focus the paper and RQs to just alpine. The value of this study is the multiseason data in the**

**same system.**

Response: Yes, we selected the best version (in your opinion) to revise the introduction section that we introduced the research in both Arctic and alpine and then focused the paper and RQs to just alpine **(Page 2 lines 13, 16; Page 3 lines 2, 5, 9, 15; Page 4 lines 4, 6, 14-17; Page 5 lines 2-8)**.

**2. The methods are unclear (why 3 sites, when are these mentioned again? Is winter vs summer sample processing associated with seasonal shift in results? Description of fumigation is confusing) and the description of the statistics is missing important details (why and how bin into seasons, and why no random factor for time?). These issues can all be fixed (I think) and a bit more effort will make this a nice paper.**

Response: Yes, the methods were thoroughly revised according to your comments. First, we introduced why 3 sites were selected and how we analysed the soil samples collected from the 3 sites, i.e., "Considering the soil spatial heterogeneity, three adjacent sites, approximately 100 m apart (centered at 32°59′ N, 103°40′ E, 3980 m a.s.l.) were selected. One site is located at the upper part of the alpine meadow, one at the middle part, and one at the lower part. Five replicates were collected from each site. The replicates from each site were 10 m apart from one another. The samples collected from the three sites (n = 15) at each sampling time were used for the statistical analyses" **(Page 6 lines 8-13)**.

Second, "3.4 Soil water content, microbial and nutrient analyses" section was divided into two sections, i.e., "3.4 Soil water content and nutrient analyses" and "3.5 Soil microbial biomass and community analyses" **(Page 7 line 15 to Page 9 line 11)**.

Finally, we rewrote the "Statistical analyses" section, i.e., "The normal distribution and homogeneity of variance of the sample datum were analyzed with SAS 9.2 software (SAS Institute Inc., 2008). The results met the basic requirements of variance analysis. Microbial and nutrient variables were analyzed to test the intra-annual differences between the growing season (i.e., data from May to October were used as a sample set; n = 90) and nongrowing season (i.e., data from November to April were used as a sample set; n = 90). Their interannual differences were also tested. Two-way ANOVA

was performed via mixed-effects model, with season and year specified as fixed effects. For the analysis of the microbial community shifts during the transition between nongrowing and growing seasons, differences in the number of bacteria, fungi, and actinomycetes between the late nongrowing season (i.e., in March) and early growing season (i.e., in May) were determined via two-way ANOVA. This procedure was performed for 2 years (2012 and 2013), and season and year specified were used as fixed effects. Pearson correlation analysis was then performed to analyze the correlation between MBC and SWC and that between MBC and DOC during the nongrowing and growing seasons. Significant results were determined at the $p < 0.05$ level, and Duncan's test was performed to analyze the significant results of the multiple comparisons to the interaction effects between season and year (SAS Institute Inc., 2008)." **(Page 10 line 10 to Page 11 line 6).**

**Specific comments by line number, with a focus on introduction and methods since the rest may change once the introduction and methods are improved.**

**Introduction: P2, l17. Edwards and Jefferies is an arctic reference, not alpine**

Response: Yes, we changed "alpine ecosystems" into "cold ecosystems" **(Page 3 line 2).**

**P3, l1, these papers show activity, but not mechanism, and not from alpine soils (which often do not freeze deeply) - perhaps remove 'alpine' and change/add a mechanistic or review ref, such as Panikov 2006 SBB, or Jefferies 2010 SBB**

Response: Yes, we changed "alpine" into "frozen", and "Panikov et al., 2006; Jefferies et al., 2010" were added into the quotation **(Page 3 line 5-6).**

**P3, l11, missing Buckeridge SBB 2013 here and possibly in next line (although this is not an alpine ref, but the study is very similar to this one despite focus on one year only) – then in line 13 the refs are a mix of alpine and Arctic, so it is not clear why a mix of refs would be used in some places and not others, and why this very similar study is not cited. P3, l17, again, mix of alpine and Arctic refs when alpine stated**

Response: Yes, "Buckeridge et al., 2013" was added, and "alpine" was revised as "Arctic and alpine" **(Page 3 lines 15-16).**

**P4, l5, missing Buckeridge et al 2010 Biogeochemistry**

Response: Yes, "Buckeridge and Grogan, 2010" was added (**Page 4 line 12**).

**P4, l7-8, the lack of summer studies is surprising, and incorrect- there are lots of studies in the summer. Perhaps be more specific - the value of this study is a multi year investigation that encompasses both summer and winter, that is rare in alpine (Edwards and Jefferies 2013 already covered this in 2 Arctic systems).**

Response: Yes, we rewrote this sentence as "However, despite ample evidence of soil microbial activity and nutrient mineralization during the winter and/or summer months in Arctic and alpine regions (Edwards et al., 2006; Schmidt et al., 2007; Miller et al., 2009; Edwards and Jefferies, 2013; Buckeridge et al., 2013), studies that explore the changes in microbial and N pools in alpine ecosystems during summer and winter across several years are few." (**Page 4 lines 13-17**).

**P4, l11-14, #3 repeats #1, and how are these RQs novel? why do we need to have this information when Brooks (1998), Lipson (2002, 2004), Edwards (2006), Larsen (2007) and Buckeridge (2010, 2013) already showed this? These RQs need to be more specific about how this particular dataset advances the field. They should also be tied to the methods and results and the alpine setting – why compare seasons and years, what questions do the authors want to address by doing this?**

Response: Yes, we rewrote the RQs as "1) What are soil microbial and available N dynamics during the growing and non-growing seasons in the alpine meadow? 2) What are interannual patterns of soil microbial and available N dynamics in the alpine meadow? 3) What environmental factors affect these dynamics? 4) What are the relationships between soil microbial biomass and available N pools in the seasonal frozen ecosystems?" (**Page 5 lines 2-7**).

**Methods: P5, l12, I do not see these 3 sites again, just the seasonal data – where are the three sites? Were these samples pooled or only one used?**

Response: The locations of the 3 sites were added, i.e., "Considering the soil spatial heterogeneity, three adjacent sites, approximately 100 m apart (centered at 32°59′ N, 103°40′ E, 3980 m a.s.l.) were selected. One site is located at the upper part of the alpine meadow, one at the middle part, and one at the lower part." (**Page 6 lines 8-10**). Fifteen

samples collected from the three sites at each sampling time were then performed together for statistical analyses (n = 15) (**Page 6 lines 11-13**).

**P5-6, what is the snow depth and timing at these sites?**

Response: Sorry, we did not measure the snow depth and timing at the three sites in 2011 to 2013. But we investigated the snow depth of the alpine meadow during the nongrowing season in 2012-2013, and the mean snow depth and timing were described in the "Site description" section, i.e., "Persistent snow cover usually occurs from late December to early April, and the mean snow depth is 16.58 cm in the study area (Xu, unpublished data, collected in 2012, 2013)" (**Page 5 lines 15-17**).

**P6, l1-3, the different treatment for winter (large roots removed) and summer (sieving 2mm) samples may explain different seasonal microbes and nutrient pool sizes - please indicate when this switch in handling occurred.**

Response: Yes, we added detailed months behind the cold periods and warm seasons, i.e., "the cold periods (i.e., November to April)" (**Page 7 line 1**) and "the warm seasons (i.e., May to October)" (**Page 7 line 4**).

**P6, l4, are the 3 subsamples analytical replicates?**

Response: NO, the 3 subsamples were analyzed for soil water content, nutrient, and microbial biomass and community, respectively.

**P6, l8, how many iButtons for each temperature measurement?**

Response: The mean daily temperatures were then calculated by the data of nine iButtons, i.e., "Three iButton data loggers were placed at each site, and mean daily temperatures were then calculated from the data of the nine loggers." (**Page 7 lines 11-12**).

**P6, l10, how was seasonal temperature calculated – by date or temperature? By date: A seasonal divide is needed – were all temp points used or were those near thaw and freeze excluded? How did the authors account for moving freeze and thaw dates across years? Or by temperature – what was the threshold, and was it based on soil or air temp?**

Response: The seasonal temperature was calculated by date, i.e., "the growing season was from 1 May to 31 October, and the nongrowing season was from 1 November to

30 April" (**Page 7 lines 12-14**). All temperature points were used for calculating.

**P6, l11 to p8, l4, this section is very confusing, for several reasons: the content does not match the order of the title, the TDN paragraph includes the description of fumigation, probably because the authors used the fumigation control for measuring TDN, and so they are confusing their operational process with the description. However, the biomass calculations were introduced first in the section, before the biomass extraction protocol, which is backwards.**

Response: Yes, we rewrote this section, and the "3.4 Soil water content, microbial and nutrient analyses" section was divided into two sections, i.e., "3.4 Soil water content and nutrient analyses" and "3.5 Soil microbial biomass and community analyses" (**Page 7 line 15 to Page 9 line 11**).

**P7, l6, Does this CFU counting follow a standard protocol? Why no reference or brief protocol when so much explanation for the dilution and fumigation method?**

Response: Yes, the CFU counting followed a standard protocol, and references were added (**Page 9 lines 2, 3, 6**).

**P8, l6-14, there are a few problems with this section: 1. mentioned above already, how seasonal binning of data was performed, also, it is not clear why the specific months were selected for community analysis; 2. The analyses of the independent variables (season and year) on the dependent variables should utilise a mixed-effects model with sample ID as a random effect to account for the lack of independence of samples across time.**

Response: Yes, we clarified the criterion of seasonal binning of data, i.e., "the growing season (i.e., data from May to October were used as a sample set; n = 90) and nongrowing season (i.e., data from November to April were used as a sample set; n = 90)" (**Page 10 lines 8-9**). We also clarified the reason why the specific months were selected for community analysis, i.e., "For analyses of the microbial community shifts during the transition between nongrowing and growing seasons" (**Page 10 lines 11-12**). Finally, the mixed-effects model was performed for the analyses of the independent variables (season and year) on the dependent variables, and new statistical results were listed in Table 1 (**Page 10 lines 10-11; Page 29**).

**Results: P8, l18-P9,l3, this passage describes a good reason why alpine and Arctic studies should be differentiated: these are not permafrost soils and they are not very cold. These mean 'freezing' soil temperatures are probably not experienced as freezing to a microbe full of osmolytes or a soil full of salts: : :.although perhaps they are during extreme lows – these extreme lows should be described, in timing, depth and frequency. Are how are freeze-thaw cycles defined? What is 'more cycles' –number and dates of FT cycles should be stated for each year.**

Response: Yes, we totally agreed with your comments, and we added the number of extreme freezing days (below −5 °C) **(Page 11 lines 6-7)**. Actually, we did not measure the frequencies of freeze–thaw cycle events, and we speculated the freeze–thaw cycle event according to the mean soil temperature (0 ℃ or thereabout). It is unreasonable to define a freeze–thaw cycle event just according to soil temperature. So, this result was deleted in the revised manuscript.

**P11, l16, please clarify a 'significantly reducing process' – soil redox measured with mV, or personal observation based on what criteria?**

Response: Yes, this sentence was revised as "Furthermore, an obviously decreasing trend of $NO_3^-$–N contents was observed during the soil thawing period (April to May)" **(Page 14 line 5)**.

**Discussion: P12, l12, again, more refs here: Brooks 1998, Edwards 2006, Larsen 2007, Buckeridge 2010.**

Response: Yes, these references were added into the revised manuscript **(Page 15 lines 1-2)**.

**P12, l16, 'temperature threshold' for what specifically? Survival, lysis? And how does the MBC decline imply high activity in cold periods? Are the authors inferring mid-winter predation?**

Response: The "temperature threshold of these cold-adapted microbial communities" was revised as "temperature threshold of the survival of these cold-adapted microbial communities" **(Page 15 line 6)**. It is unreasonable to inferring that the decline of MBC imply high activity in cold periods. So, the sentence "and these communities retained their high activity in alpine soils during the cold periods" was deleted in the revised

manuscript (**Page 15 lines 7-8**).

**P13, l1, 'even though' does not make sense here.**

Response: Yes, the sentence "even though the N uptakes of plants were degraded" was deleted in the revised manuscript (**Page 15 line 9**).

**P14, l9-10, the second half of this sentence is not useful**

Response: Yes, the sentence "which might contribute to the seasonal dynamics of the microbial biomass" was deleted in the revised manuscript (**Page 16 line 18**).

**P14, l18 & P16, 8, the frequency and number of freeze-thaw cycles was not stated in the results**

Response: Yes, as we did not measure the frequencies of freeze–thaw cycle events, some similar literatures were cited in the revised manuscript. The sentence was revised as "Notably, a warmer and drier nongrowing season was observed in 2011–2012 than that in 2012–2013, which might accompanied with more frequent freeze–thaw cycles during the early period of this season (Mellander et al., 2007; Henry, 2008)" (**Page 18 lines 16-18**).

**P16, l1-5, is this discussion based on gravimetric water content? Can the authors comment on why gravimetric content would correlate with non-growing season biomass if this water was frozen and unavailable? Fig.2 and associated data: are these values for gravimetric water content? How meaningful are the conclusions drawn from water pool sizes and correlations if the frozen soil water is not removed from the calculations?**

Response: Yes, we agree with your comments. Actually, the discussion was based on the gravimetric water content during the growing season. Furthermore, low correlation ($r = 0.35$) between MBC and SWC was observed during the nongrowing season. We thought the frozen soil water might be correlated with the MBC during the soil thawing period.

**Fig.4 the lowercase letters represent the post-hoc test for which effect? The interaction? Fig.8, again not clear which main effect test the post-hoc letters are representing.**

Response: In Fig.4 and Fig.8, the lowercase letters represented the post-hoc test for the

interaction effects between season and year, and we clarified it in the revised manuscript **(Page 10 lines 17-18; Page 33 lines 11-12; Page 36 lines 15-16)**.

**Responses to Anonymous Referee #4**

**General comments**

**This paper describes intra-annual and inter-annual patterns in soil nutrient availability (inorganic and organic N) as well as microbial biomass and community structure in alpine tundra. The investigators sampled soils monthly over a 3 year period, including both the frozen and unfrozen periods. This is an impressive data set and I'm not aware of another published data set that is nearly as comprehensive. For this reason alone I encourage the authors to continue to work towards the publication of this data set. There are some aspects of both the methods and the interpretation of the results which I question and these aspects in particular require more attention by the authors before publication of this paper. See more specific comments below.**

Response: We thank referee for the helpful comments. After discussing with co-authors, we thoroughly revised the manuscript.

**Specific comments**

**Referencing: Some of the references are inappropriate. Specifically, there are many citations which are used to support statements about alpine systems which were not conducted in alpine ecosystems (E.g. Page 2 line 17 and Page 4 line 8 Edwards and Jefferies, Page 3 line 6 Buckeridge and Grogan, Page 15 line 4 Henry and Jefferies). Some references are missing (Page 14 line 3: reference for Alaskan tundra is missing) and others did not examine the phenomena they are used to support (e.g. Edwards and Jefferies did not examine the survival of microorganisms surviving in thin water films (Page 3 line 1).**

Response: Yes, we carefully revised these inappropriate references one by one in the new manuscript **(Page 2 lines 12, 15, 18; Page 3 lines 3, 4, 7, 12; Page 4 lines 3, 10-13)**.

**The methods are lacking some necessary details. The description of the 3 sites were**

**vague: The sites are described as being at the "top middle and bottom of the meadow". Were there elevational differences between the sites? How far is the distance between them?**

Response: Yes, the details of the 3 sites were added, i.e., "Considering the soil spatial heterogeneity, three adjacent sites approximately 100 m apart (centered at 32°59′ N, 103°40′ E, 3980 m a.s.l.) were sampled, namely located at the upper, middle, and lower part of the alpine meadow. Five replicates at each site were collected, and the replicates from each site were 10 m apart from each other. Fifteen samples collected from the three sites at each sampling time were then performed together for statistical analyses (n=15)." (**Page 6 lines 5-9**).

**Further, were the soils collected in the winter kept frozen into analysis?**

Response: Yes, the soil samples collected in the winter were stored at 0 ℃ before analysis, and all the samples were processed at the laboratory of Chengdu Institute of Biology, CAS, within two days of sampling (**Page 7 lines 1-2**).

**Finally, was TDN measured only after chloroform fumigation? This is how it is described, but then it would be impossible to measure MBC and MBN.**

Response: No, different subsamples were used for the determinations of TDN, MBC and MBN. We rewrote this section, and the "3.4 Soil water content, microbial and nutrient analyses" section was divided into two sections, i.e., "3.4 Soil water content and nutrient analyses" and "3.5 Soil microbial biomass and community analyses" (**Page 7 line 15 to Page 9 line 11**).

**It would also be good to report days below -5C rather than just below 0C: -5C is often reported as when microbial activity significantly slows.**

Response: Yes, we added the results of the number of days below -5 ℃ in the revised manuscript (**Page 11 lines 12-13**).

**I also question the methods used to determine changes in microbial community structure. The authors used total colony forming units of bacteria, fungi and actinomycetes using a plate dilution method. However, this only allows culturable bacteria to be counted. Further, they were all incubated at 25C regardless of season, when the winter samples likely should have been incubated at colder**

**temperatures. Also, how were these #s compared over time? The results state which dates are significantly different from each other – were they pairwise comparisons? If the authors plan to use these methods to describe microbial community structure I would like to see citations indicating they are appropriate, as well as further description of the limitations of these methods.**

Response: Actually, the dilution-plate method can be used to counting the CFU of bacteria, fungi, and actinomycetes by different selective mediums, i.e., beef extract peptone agar, Sabouraud dextrose agar, and Gause synthetic agar medium for the cultivation of bacteria, fungi, and actinomycetes, respectively (Li, 1996; Igbinosa, 2015) **(Page 9 lines 6-8)**. We thought if the cultivation temperature was too low, the visible microbial colony might hard to forming. So we referred to the methods of Li (1996), and measured the CFUs of bacteria, fungi, and actinomycetes.

For the analysis of the microbial community shifts during the transition between nongrowing and growing seasons, differences in the number of bacteria, fungi, and actinomycetes between the late nongrowing season (i.e., in March) and early growing season (i.e., in May) were determined via two-way ANOVA. This procedure was performed for 2 years (2012 and 2013), and season and year specified were used as fixed effects **(Page 10 lines 16-18 to Page 11 lines 1-2)**.

**Statistics: Because the same sites/plots were sampled repeatedly, a repeated measures ANOVA would be more appropriate than the 2-way ANOVA. Further, the description of the Pearson correlation analysis is not clear. I would like to see more of the results for this correlation described than just the r2 (Table 2).**

Response: We thought the analyses of the independent variables (season and year) on the dependent variables should utilize a mixed-effects model with sample ID as a random effect to account for the lack of independence of samples across time. So, the mixed-effects model was performed for the analyses of the independent variables (season and year) on the dependent variables **(Page 10 lines 15-16)**, and new statistical results were listed in Table 1 **(Page 29-30 )**. Further, we revised the description of the Pearson correlation analysis as "Pearson correlation analysis was then performed to analyze the correlation between MBC and SWC and that between MBC and DOC

during the nongrowing and growing seasons." **(Page 11 lines 2-4)**. In Table 2, information on *r* and *p* values was listed, we thought it was enough to describe the results of the correlation analysis.

**Also, throughout the results section I would like to see the actual statistics stated rather than just p<0.05. Finally, is it possible to define a "peak" time for MBN or DON in the season when MBN did not vary seasonally? (Page 9 line 5).**

Response: Yes, we added the actual statistics results in the two-way ANOVA analysis throughout the results section **(Page 12 lines 5, 16-17; Page 13 lines 12-13; Page 14 lines 2-3, 11-14, 18; Page 15 line 1)**, but the description of "*p*<0.05" was retained in the sections of the multiple comparison and Pearson correlation analysis. Finally, it is possible to define a "peak" time for MBN or DON according to their monthly values, and the MBN or DON had no significant seasonal differences just compared between growing and nongrowing seasons.

**Interpretation: Some of the interpretation of the results goes beyond what the results actually indicate. For example (Page 12 line 17) High microbial biomass does not mean there is high activity.**

Response: Yes, the sentence "and these communities retained their high activity in alpine soils during the cold periods" was deleted in the revised manuscript **(Page 15 lines 16-17)**.

**Also see a reference to activity on page 14 line 16: this study did not contain any tests of microbial activity.**

Response: Yes, "Lipson et al., 1999; Matthew Robson et al., 2010" were added **(Page 17 line 17)**.

**Other conclusions require further elaboration. For example, the section on page 13 line 16 needs elaboration – Why would the decrease in MBC at thaw be related to the higher productivity and SOM in this site compared with others?**

Response: Actually, we did not get the conclusion that the decrease in MBC at thaw be related to the higher productivity and SOM in this site compared with others. But, we inferred that available C and N were relatively sufficient and might not restrict the microbial activity during the winter-spring transition, and this phenomenon may be

closely related to the high plant community productivity and SOM in our study compared with others.

**Finally, there isn't direct support for many of the overall conclusions of the paper – this study can describe correlations, but not the types of conclusions described (e.g. soil microorganisms play a crucial role in accumulation of inorganic N pools)**

Response: Yes, we revised it as "Furthermore, the soil microorganism not only has a close correlation with the accumulation of inorganic N pools but also is an important soil organic N pool itself." **(Page 21 line 1-3)**

**Technical comments**

**The paper could use a thorough editing for English grammar: E.g. Community compositions should be community composition (Page 1 line 16) E.g. Change "Consistently increasing trends of MBC" to "Trends of consistently increasing MBC" E.g. Substrate transports should be substrate transport (Page 2 line 4)?**

Response: Yes, we revised them one by one according to your comments **(Page 1 lines 16, 18-19; Page 3 line 7**), and the revised manuscript has been sent to a professional language editing company for the language modification.

Thank you again for your suggestion!

Best regards!

Bo Xu

[revised manuscript text omitted]

F̶i̶g̶.̶ ̶3̶.̶ ̶D̶y̶n̶a̶m̶i̶c̶s̶ ̶o̶f̶ ̶m̶i̶c̶r̶o̶b̶i̶a̶l̶ ̶b̶i̶o̶m̶a̶s̶s̶ ̶C̶ ̶(̶A̶)̶ ̶a̶n̶d̶ ̶N̶ ̶(̶B̶)̶ ̶i̶n̶ ̶s̶o̶i̶l̶s̶ ̶o̶f̶ ̶t̶h̶e̶ ̶a̶l̶p̶i̶n̶e̶ ̶m̶e̶a̶d̶o̶w̶ ̶f̶r̶o̶m̶ ̶A̶p̶r̶i̶l̶ ̶2̶0̶1̶1̶ ̶t̶o̶ ̶O̶c̶t̶o̶b̶e̶r̶ ̶2̶0̶1̶3̶

(̶m̶e̶a̶n̶ ̶±̶s̶.̶e̶.̶;̶ ̶$̶n̶ ̶=̶ ̶1̶5̶$̶)̶.̶

[Figure]

Fig. 4. Dynamics of microbial biomass C and N (A and B; mean ± s.e.; $n = 15$), and their seasonal and interannual

differences (C; mean ±s.e.; $n = 90$) from April 2011 to October 2013 (mean ±s.e.; $n = 90$). The sampling time was on

the 15th day of each month during the growing season from May to October, and during the nongrowing season from

November to April next year. Seasons and years were compared using two-way ANOVA, and different lowercase letters

indicate significant differences of the interaction effects between season and year determined via Duncan test ($p < 0.05$).

Fig. 4. Changes in microbial biomass C (MBC) and N (MBN) in the growing and non-growing seasons from 2011 to 2013 (mean ±s.e.; $n$ =90). The sampling time was on the 15th day of each month during the growing season from May to October, and during the non-growing season from November to April next year. Seasons and years were compared using two-way ANOVA, and different lowercase letters indicate significant differences ($p < 0.05$) determined via Duncan test.

[Figure]

Fig. 5. Changes in the number of bacteria (A), fungi (B), and actinomycetes (C) during the transition between freezing and thawing periods (mean ±s.e.; $n = 15$). The sampling time during the freezing period was on 15 March and during the thawing period was on 15 May each year. Different lowercase letters indicate significant differences of the interaction

effects between season and year  according to two-way ANOVA (p < 0.05).

[Figure]

Fig. 6. Dynamics of dissolved organic C (A; mean ± s.e.; $n = 15$) and its seasonal and interannual  differences

(B; mean ± s.e.; $n = 90$) from 2011 to 2013.

[Figure]

Fig. 7. Dynamics of $NH_4^+$–N(A), $NO_3^-$–N(B), and DON(C) in soils of the alpine meadow from April 2011 to October 2013 (mean ± s.e.; $n = 15$).

[Figure]

Fig. 8. Changes in $NH_4^+$–N, $NO_3^-$–N, and DON of growing and nongrowing seasons from 2011 to 2013 (mean $\pm$ s.e.; $n$ =90). The sampling time was on the 15th day of each month from May to October during the growing season and during the nongrowing season from November to April next year. Seasonal and interannual differences were compared using two-way ANOVA. Different lowercase letters indicate significant differences of the interaction effects between season and year  determined via Duncan test ($p < 0.05$).

---

## Author Response (AR2)

Dear Prof. Michael Weintraub,

Thank you very much for your helpful comment to our manuscript, and we have carefully and thoroughly revised the manuscript according to your and referees' comments. The detailed responses to the comments are as follows.

Response to Anonymous Referee #1

**The authors have made substantial progress. Here I note the some comments that are, in my view, still not addressed adequately.**

Response: We thank referee for the helpful comments. After discussing with co-authors, we thoroughly revised the manuscript and listed in supplement.

**Correction factors: it's the 0.45 and 0.54 (now page 8, L8). I recommend not correcting these numbers using these factors.**

Response: Thank you for your comment. But, we thought correction factors of 0.45 for C and 0.54 for N were reasonable for the determination of MBC and MBN.

**There is still no reason given for the pseudoreplicative design (ie three non-independent sets of 5 samples that are then treated as independent). It's not ideal, but there are often acceptable reasons for it and so it's not a fatal flaw in the study. However, I'd like the know the authors' thinking here. The way I see it there are two possibilities: (1) logistically it was not possible to randomly sample across the entire area (meadow or meadow system) of interest, but the authors wanted 15 samples so this is how they were able to achieve that. (2) they were originally planning on comparing the sites, but there were not a lot of interesting differences so they pooled them together in order to achieve**

**a better overall understanding of seasonal trends, which were perhaps not visible with n = 5 at each site. This was a point brought up by multiple reviewers and the authors need to provide a justification on why such a design was used.**

Response: Thank you for your comments. Actually, we have explained why we selected 3 adjacent sites that because of the consideration of the soil spatial heterogeneity in the alpine meadow. Furthermore, the 15 samples at each sampling time were independent, because they were randomly collected at different locations **(Page 6 lines 2-5)**.

**Figures: having the bar graphs below the line graphs further emphasizes that the information is redundant. I'll leave it up to authors and editor, but I still recommend removing the bar graphs.**

Response: Thank you for your comment. But, we did not think having the bar graphs below the line graphs further emphasizes that the information is redundant. Because, they showed different information, i.e., the line graphs showed detailed information on intra- and interannual patterns of microbial and nutrient dynamics; the bar graphs showed significant differences in microbial biomass and nutrients between seasons and years, and their interaction effects. Thus, these figures were indispensable for our study.

**Inclusion of data in a publicly available repository: the authors did not address this request. I still recommend it.**

Response: Yes, The data set related to this study has been provided as a supplement.

**Re: TDN vs. MBN numbers. Some ok points made in the reviewer response, but it's not in the paper. It warrants discussion in the paper.**

Response: Thank you for your comment. Actually, the important points made in the response have been stated in the discussion section **(Page 17 lines 1-5; Page 18 lines 8-13)**.

**Photograph and map are not the same thing. It helps to see what the**

**ecosystem looks like on the ground. I still recommend this, even if in the supp materials.**

Response: Sorry, we did not have appropriate photographs to show the alpine meadow.

**A comment from another reviewer about unsubstantiated statements in the discussion is still relevant: "Finally, there isn't direct support for many of the overall conclusions of the paper – this study can describe correlations, but not the types of conclusions described (e.g. soil microorganisms play a crucial role in accumulation of inorganic N pools)" Some examples of these statements:**

**"This period of active microbial activity and N mineralization benefited from substrates conducive for microbial growth, particularly those supplied by the fresh plant litter inputs in autumn."**

Response: Thank you for your comment. Here, we did not make any conclusion, but only cited some relevant results of previous researches. We revised "benefited from" as "might benefit from" (**Page 14 line 8**).

**"Snow melting during this period is an important source of NH4 +–N" This is only true with high deposition. Not sure if this region is susceptible to that.**

Response: Thank you for your comment. Actually, this sentence was also a citation of previous research, which is a potential reason why "A trend of increasing $NH_4^+$–N content was found during the early soil thaw". We revised "is an" as "may be an" (**Page 16 line 14**).

**"At the start of the growing season, NH4 +–N content sharply decreased, partly because alpine meadow plants prefer NH4 +–N" maybe change partly to possibly**

Response: Yes, "partly" has been changed into "possibly" (**Page 16 line 6, 15; Page 18 line 13**).

**"the seasonal dynamics of different available N pools showed**

**significant complementarity with the nutrient supply process and play a crucial role in maintaining abundant biodiversity of alpine meadow ecosystem"**

Response: Yes, we have revised "and play" as "and will play", and some relevant references were added (**Page 17 line 6**).

**"However, they showed a divergent interannual pattern during the growing season, partly because of the plant and microbe uptakes and leaching effects." same use of partly. maybe authors mean to use possibly in these cases.**

Response: Yes, "partly" was changed into "possibly".

**"According to our monitoring results, soil temperature and water condition are the primary environmental factors driving the seasonal and interannual dynamics of soil microbial biomass and available N pools." I would probably leave this out. It's something most would accept but at the same time, is not really shown by this study, which does not address mechanisms.**

Response: Thank you for your comment. But, we thought this conclusion could be made according to our results (**e. g., Figures 2, 3; Tables 1, 2**) that "soil temperature and water condition are the primary environmental factors driving the seasonal and interannual dynamics of soil microbial biomass and available N pools".

**Finally, I would still axe the mentions of climate change from the conclusion. The contributions of this paper are on seasonal trends, not climate change. It's not a huge change. It's fine to mention it where it is mentioned just before the conclusion, but it should not be emphasized as the final statement in the paper. The authors justification in the response that because temperature and moisture appear to correlate with the other measured variables, climate change is important is not convincing. Temperature and moisture are always**

**important for microbial processes, and it's a leap to then suggest that this paper provides particular insight on how this relationship will change with climate.**

Response: Thank you for your comment. But, we still thought it was necessary to mention the climate change in the conclusion. Because temperature change and uneven distribution of precipitation are two important aspects of climate change. Furthermore, the alpine ecosystems are sensitive to the future climate change.

**minor:**

**"An obviously trend of increasing NH4 + –N content was found during the early soil thaw" this revised sentence does not make sense**

Response: Thank you for your comment. But we thought this sentence was important, because it showed a pulse phenomenon of $NH_4^+$–N during the late nongrowing season, which might play a crucial role in nutrient supplies for plants during the early growing season.

Response to Anonymous Referee #2

**I think that thanks to the suggestions of the referees the paper has considerably improved. However I suggest to the authors some further changes, listed below:**

**Pag. 27 line 7: change avilable into available. See also pag 27 line 10.**

Response: Yes, "avilable" was changed into "available" (**Page 4 lines 3, 6**).

**Pag 28 lines 6-7: Did you add also the soil classification according to the Soil Taxonomy (Silty Loam Inceptisol)? If yes please add the proper reference: Soil Survey Staff. 2014. Keys to Soil Taxonomy, 12th ed. USDA-Natural Resources Conservation Service, Washington, DC.**

Response: Yes, the reference that "Soil Survey Staff: Keys to Soil Taxonomy, 12th ed. USDA-Natural Resources Conservation Service, Washington, DC., 2014." was added (**Page 5 line 18; Page 26 lines 6-7**).

**Pag 28 lines 10-11: Please specify the elevation of the 3 sites**

Response: Sorry, we only measured the elevation of the center site, because the 3 sites were adjacent, and their differences in elevation were very small.

**Pag 29 line 1: add O horizon before living plant roots and litter**

Response: Yes, "O horizon" was added before "living plant roots and litter" **(Page 6 line 11)**.

**Pag 35 line 14: sorry I don't understand this sentence. What do you mean with "from each other"**

Response: "from each other" means that significant differences compared with each other. We have deleted "from each other" in the revised manuscript **(Page 12 line 12)**.

**Pag 35 line 18: What do you mean woth the term those**

Response: Sorry, we could not find "those" in Page 35 line 18, we were not sure which "those" in the manuscript did you refer.

**Pag 37 line 10: delete early**

Response: Sorry, we could not find "early" in Page 37 line 10, we were not sure which "early" in the manuscript did you refer.

**Pag 38 line 18: what do you mean with a-1**

Response: The "$a^{-1}$" in the units for biomass refers to per year.

**Pag 40 line 1: Why obviously?**

Response: "obviously" was deleted **(Page 13 line 10; Page 16 line 11)**.

**Pag 43 line 5: Delete monitoring**

Response: Yes, "monitoring" was deleted **(Page 18 line 5; Page 19 line 11)**.

Response to Anonymous Referee #4

**This paper describes both seasonal and interannual variability in soil microbial biomass and soil available N in alpine tundra with monthly**

**resolution over a 3 year time period. This is an impressive data set which is worthy of publication. I reviewed an earlier version of this manuscript and made a number of suggestions. The writing is much improved but still requires further work. Further, some of my previous comments have not been dealt with to my satisfaction, as described below.**

Response: We thank referee for the helpful comments. After discussing with co-authors, we thoroughly revised the manuscript and listed in supplement.

**In the previous version I had questions about the MBC/MBN methods as well as the statistics. Neither of these have been dealt with satisfactorily. The methods for determining TDN are still not clear. The procedure for TDN is described on page 7 line 11-16. Line 9-11 describes the chloroform fumigation procedure which is not the methods for TDN but are part of the methods for MBN. TDN is determined on both fumigated and non-fumigated samples (the non-fumigated sample analysis is currently not described in the paper) and the difference is MBN. The chloroform fumigation methods could be moved to the paragraph beginning on page 8 line 6.**

Response: Yes, the methods for determining TDN, MBC and MBN were rewrote according to your suggestion (**Page 7 lines 8-16; Page 8 lines 7-17**).

**For the statistics, the response to reviewer 4 indicates that sample ID was included in the model to account for the lack of independence of samples across time. However, the stastistical analysis section of the manuscript does not describe any inclusion of sample ID in the model. Please clarify in the manuscript. Also, all F values throughout the manuscript should include the degrees of freedom.**

Response: Thank you for your comment. Actually, we have described the

sample ID before the description of the mixed-effects model, i.e., "Microbial and nutrient variables were analyzed to test the intra-annual differences between the growing season (i.e., data from May to October were used as a sample set; n = 90) and nongrowing season (i.e., data from November to April were used as a sample set; n = 90)" **(Page 9 lines 15-18)**.

Yes, the degrees of freedom (df) were added behind the $F$ values throughout the revised manuscript **(Page 11 lines 5, 15-16; Page 12 lines 11-12; Page 13 lines 1-2, 8-9, 14-15)**.

**Some parts of the result section are also not clear. For example, when describing DOC patterns, the seasons are described as being not significantly different from each other (page 11 line 18) and also significantly different (page 12 line 4).**

Response: Thank you for your comment. Although, "the seasonal dynamics of DOC had no significant difference", the interaction effects of DOC between season and year were significantly different. Thus, "the DOC contents during the nongrowing season in 2011–2012 (174.27 mg kg$^{-1}$ ± 32.59 mg kg$^{-1}$) and growing season in 2012–2013 (170.85 mg kg$^{-1}$ ±41.19 mg kg$^{-1}$)" showed the result that "were significantly lower than that in other seasons".

**Also, the results section could be reduced – e.g. The two sentences from page 10 line 17 to page 11 line 1 say the same thing.**

Response: Thank you for your comment. Actually, the two sentences described different things, i.e., the first sentence showed that "the MBC values in the nongrowing seasons were consistently higher than those in the growing seasons"; the second sentence showed that "the mean MBC value during the nongrowing season in 2012–2013" was the highest among different seasons.

**Finally, just as a suggestion, figure 3 could also be presented with only**

**part A and shading to indicate the different season – this way the data is only presented once rather than repeated in both parts of the figure. The same change could be applied to the other figures. The two types of presentations are presented in the same figure sometimes (Figure 3) and as separate figures in others (Figure 7 and 8).**

Response: Thank you for your comment. But, we thought these figures were indispensable for our study. Because, they showed different information, i.e., the line graphs showed detailed information on intra- and interannual patterns of microbial and nutrient dynamics; the bar graphs showed significant differences in microbial biomass and nutrients between seasons and years, and their interaction effects.

**In the discussion, the authors still need to be cautious about implying causality for some of the patterns they have measured. Some examples are below:**

**Page 13 Line 14 – indicate that the mechanism for the increase in soil MBC and available N are speculative. Also, the conclusion of this paragraph describes an accumulation of organic N which is not described in the remainder of the paragraph.**

Response: Yes, this sentence was deleted (**Page 14 lines 12-14**).

**Page 15 Line 15 – Only describe what you have evidence for – e.g. an increase in microbial biomass and not activity**

Response: Yes, the microbial activity was revised as "microbial biomass" (**Page 16 line 9**).

**And lastly, a few clarifications are required in the discussion:**

**Page 15 line 9 – what do you mean by the "number of bacteria" increased just after thaw? The previous paragraph describes a crash in microbial biomass. Do you mean the proportion of bacteria to fungi? The number of bacterial phenotypes?**

Response: Here, the "number of bacteria" mean the number of CFU of

bacteria.

**Page 15 line 2-5 The units for soil organic matter need an area (per m2?) for this comparison to be relevant. Is this sentence implying that the SOM in tundra is limiting to microbial growth in some circumstances?**

Response: Thank you for your comment. But, we thought the units for soil organic matter did not need an area, because we just need compare the differences of soil organic matter contents among different alpine meadows. Furthermore, we also did not know the values of SOM per square meter ($m^2$) from the references. According to this sentence, we knew that the SOM contents in the alpine meadows of the Qinghai-Tibet Plateau were relatively higher than that in other alpine meadows. It was implied that the available C was relatively sufficient in this region and "might not restrict microbial activity during the winter–spring transition" (**Page 11 lines 9-10**).

**The writing in this version is much improved over the last version. However, the writing still needs to be improved for clarity and grammar. A number of examples follow:**

**Page 1 Line 10 – replace "occurs seasonally" with "varies seasonally" as the activity occurs all the time**

Response: Yes, "occurs seasonally" was replaced with "varies seasonally" (**Page 1 line 10**).

**Page 1 Line 15 – replace "Topsoil samples were" with "Soil was" and "and were analyzed" with "and analyzed"**

Response: Yes, "Topsoil samples were" and "and were analyzed" were replaced with "Soil was" and "and analyzed", respectively (**Page 1 line 15**).

**Page 1 Line 17 – replace "was measured" with "as measured"**

Response: Yes,"was measured" was replaced with "as measured" (**Page 1 line 17**).

**Page 1 Line 18 – replace "the number of" with "the proportion of"**

Response: Yes,"the number of" was replaced with "the proportion of" (**Page 1 line 18**).

**Page 2 Line 1 – replace "induced by soil temperatures" with "induced by changes in soil temperatures"**

Response: Yes,"induced by soil temperatures" was replaced with "induced by changes in soil temperatures" (**Page 2 line 2**).

**Page 3 Line 11 – delete "apparently"**

Response: Yes, "apparently" was deleted (**Page 3 line 11**).

**Page 4 Line 5 Delete "In alpine systems" as it is repeated later in the sentence**

Response: Yes,"In alpine systems" was deleted (**Page 4 line 5**).

**Page 5 line 8 Delete "and"**

Response: Yes,"and" was deleted (**Page 5 line 8**).

**Page 6 line 10 – Delete "and 15 soil samples… time" as this information is repeated from a few sentences earlier**

Response: Yes,"and 15 soil samples… time" was deleted (**Page 6 line 10**).

**Page 9 line 4 – "analysis of variance" not "variance analysis"**

Response: Yes,"variance analysis" was revised as "analysis of variance" (**Page 9 line 15**).

**Page 14 line 17 – what does the a refer to in the units for biomass? Per year?**

Response: Yes, the "$a^{-1}$" in the units for biomass means "per year" **Similarly, the use of appropriate references is also much improved. A few more which need to be changed are:**

**Page 3, line 3: Mikan is not an alpine reference**

Response: Yes, "Mikan et al., 2002" was deleted and two relevant alpine references were added, i.e., "Brooks et al., 1996; Jefferies et al., 2010" **(Page 3 line 3)**.

**Page 16 line 13 – Qin and Petechey/Gaston are general biodiversity – ecosystem functioning researchers and not appropriate for use in this sentence.**

Response: Yes, "Qin et al., 2003; Petchey and Gaston, 2006" were deleted, and two appropriate references were added, i.e., "Kahmen et al., 2006; Ashton et al., 2010" **(Page 17 lines 7-8)**.

Best regards!
Bo Xu

[revised manuscript text omitted]

Note: ns, no significant difference; **, $p < 0.01$.

**Figure legends**

[Figure]

Fig. 1. Location of the study site

[Figure]

Fig. 2. Mean daily soil temperature in the alpine meadow from April 2011 to October 2013. Thermochron iButton data

10    loggers were placed at 10 cm soil depth to obtain automatic readings every 1 h, and the mean daily soil temperature was

calculated every day.

[Figure]

Fig. 3. Dynamics of soil water content (A; mean ±s.e.; $n = 15$) and its seasonal and interannual differences (B; mean ± s.e.; $n = 90$) from 2011 to 2013.

[Figure]

Fig. 4. Dynamics of microbial biomass C and N (A and B; mean ± s.e.; $n = 15$), and their seasonal and interannual differences (C; mean ± s.e.; $n = 90$) from April 2011 to October 2013 (mean ± s.e.; $n = 90$). The sampling time was on the 15th day of each month during the growing season from May to October, and during the nongrowing season from November to April next year. Seasons and years were compared using two-way ANOVA, and different lowercase letters indicate significant differences of the interaction effects between season and year determined via Duncan test ($p < 0.05$).

Fig. 5. Changes in the number of bacteria (A), fungi (B), and actinomycetes (C) during the transition between freezing and thawing periods (mean ± s.e.; $n = 15$). The sampling time during the freezing period was on 15 March and during the thawing period was on 15 May each year. Different lowercase letters indicate significant differences of the interaction effects between season and year according to two-way ANOVA ($p < 0.05$).

[Figure]

Fig. 6. Dynamics of dissolved organic C (A; mean ±s.e.; $n = 15$) and its seasonal and interannual differences (B; mean ±s.e.; $n = 90$) from 2011 to 2013.

[Figure]

Fig. 7. Dynamics of NH$_4^+$–N(A), NO$_3^-$–N(B), and DON(C) in soils of the alpine meadow from April 2011 to October 2013 (mean ± s.e.; $n = 15$).

[Figure]

Fig. 8. Changes in $NH_4^+$–N, $NO_3^-$–N, and DON of growing and nongrowing seasons from 2011 to 2013 (mean $\pm$ s.e.; $n = 90$). The sampling time was on the 15th day of each month from May to October during the growing season and during the nongrowing season from November to April next year. Seasonal and interannual differences were compared using two-way ANOVA. Different lowercase letters indicate significant differences of the interaction effects between season and year determined via Duncan test ($p < 0.05$).

---

## Author Response (AR3)

Dear Prof. Michael Weintraub,

First of all, we are sorry for the ignoring of some important points made by the reviewers in the last round of review. Thank you very much for your helpful comments to our manuscript, and we have thoroughly revised the manuscript after carefully reconsidering every reviewers' comment. Furthermore, we have sent our paper to a professional language editing company to the language modification. The detailed responses to the comments are as follows:

Response to Anonymous Referee #1

**The authors have made substantial progress. Here I note the some comments that are, in my view, still not addressed adequately.**

Response: We thank referee for the helpful comments. After discussing with co-authors, we thoroughly revised the manuscript and listed in supplement.

**Correction factors: it's the 0.45 and 0.54 (now page 8, L8). I recommend not correcting these numbers using these factors.**

Response: Thank you for your comment. But, we thought correction factors of 0.45 for C and 0.54 for N were reasonable for the determination of MBC and MBN. First, using these extraction efficiency correction factors to correct the result was an important step in the MBC and MBN determinations according to the *Principles and Methods of Soil Microbial Research* written by Lin in 2010. Second, our previous research in the same study area found that it was necessary to using the extraction efficiency correction factors to correct the results for the determination of MBC and MBN (i.e., Wang J., Xu B., Wu Y., Gao J., and Shi F.: Flower litters of

alpine plants affect soil nitrogen and phosphorus rapidly in the eastern Tibetan Plateau. Biogeosciences, 13, 5619–5631, 2016.)

**There is still no reason given for the pseudoreplicative design (ie three non-independent sets of 5 samples that are then treated as independent). It's not ideal, but there are often acceptable reasons for it and so it's not a fatal flaw in the study. However, I'd like the know the authors' thinking here. The way I see it there are two possibilities: (1) logistically it was not possible to randomly sample across the entire area (meadow or meadow system) of interest, but the authors wanted 15 samples so this is how they were able to achieve that. (2) they were originally planning on comparing the sites, but there were not a lot of interesting differences so they pooled them together in order to achieve a better overall understanding of seasonal trends, which were perhaps not visible with n = 5 at each site. This was a point brought up by multiple reviewers and the authors need to provide a justification on why such a design was used.**

Response: Thank you for your comments. But, we did not thought it was reasonable that judging the 15 samples collected from 3 sites or locations were non-independent. Because, as you said, it was not possible to randomly sample across the entire alpine meadow. Thus, given the soil spatial heterogeneity in the alpine meadow, 3 independent sites or locations (i.e., located at the upper part, middle part, and lower part of the meadow) were randomly selected at each sampling time, and 5 samples were then randomly collected at each sites **(Page 6 lines 1-4)**. Furthermore, we agreed with that the sample sets of season and year were dependent because they were repeatedly collected at different time in the same alpine meadow. It was the pseudo-replication if we treated the dependent samples as independent samples. However, we thought it was acceptable to use mixed-effects model to analyze the dependent variations (i.e., season and year) to

understand the seasonal and interannual dynamics of soil microbial biomass and available nitrogen in the alpine meadow across 3 years.

**Figures: having the bar graphs below the line graphs further emphasizes that the information is redundant. I'll leave it up to authors and editor, but I still recommend removing the bar graphs.**

Response: Thank you for your comment. But, we did not think having the bar graphs below the line graphs further emphasizes that the information is redundant. Because they showed different information, i.e., the line graphs showed detailed information on intra- and interannual patterns of microbial and nutrient dynamics; the bar graphs showed significant differences in microbial biomass and nutrients between seasons and years, and their interaction effects. Thus, these figures were indispensable for our study.

**Inclusion of data in a publicly available repository: the authors did not address this request. I still recommend it.**

Response: Yes, The data set related to this study has been provided as a supplement.

**Re: TDN vs. MBN numbers. Some ok points made in the reviewer response, but it's not in the paper. It warrants discussion in the paper.**

Response: Thank you for your comment. Actually, the important points made in the response have been stated in the discussion section (**Page 16 lines 3-6, 13-17**).

**Photograph and map are not the same thing. It helps to see what the ecosystem looks like on the ground. I still recommend this, even if in the supp materials.**

Response: Sorry, we did not have appropriate photographs to show the alpine meadow.

**A comment from another reviewer about unsubstantiated statements in the discussion is still relevant: "Finally, there isn't direct support for many of the overall conclusions of the paper – this study can**

**describe correlations, but not the types of conclusions described (e.g. soil microorganisms play a crucial role in accumulation of inorganic N pools)" Some examples of these statements:**

**"This period of active microbial activity and N mineralization benefited from substrates conducive for microbial growth, particularly those supplied by the fresh plant litter inputs in autumn."**

Response: Thank you for your comment. Here, we did not make any conclusion, but only cited some relevant results of previous researches in the similar regions. Moreover, they were potential reasons why "the soil MBC and available N pools both increased at the beginning of the early nongrowing season". We revised "benefited from" as "might benefit from" **(Page 14 line 2)**.

**"Snow melting during this period is an important source of NH4 +–N" This is only true with high deposition. Not sure if this region is susceptible to that.**

Response: Thank you for your comment. Actually, this sentence was also a citation of previous research, which is a potential reason why "An obviously increased trend of $NH_4^+$–N content was found during the early soil thawing". And we revised it as "On the other hand, snow melting during this period may be an important source of $NH_4^+$–N" **(Page 16 line 6)**.

**"At the start of the growing season, NH4 +–N content sharply decreased, partly because alpine meadow plants prefer NH4 +–N" maybe change partly to possibly**

Response: Yes, "partly" has been changed into "possibly" **(Page 16 line 7)**.

**"the seasonal dynamics of different available N pools showed significant complementarity with the nutrient supply process and play a crucial role in maintaining abundant biodiversity of alpine meadow ecosystem"**

Response: Yes, "and play a crucial role in maintaining abundant biodiversity of alpine meadow ecosystem" was deleted (**Page 16 lines 16-17**).

**"However, they showed a divergent interannual pattern during the growing season, partly because of the plant and microbe uptakes and leaching effects." same use of partly. maybe authors mean to use possibly in these cases.**

Response: Yes, "partly" was changed into "possibly" (**Page 18 lines 4**).

**"According to our monitoring results, soil temperature and water condition are the primary environmental factors driving the seasonal and interannual dynamics of soil microbial biomass and available N pools." I would probably leave this out. It's something most would accept but at the same time, is not really shown by this study, which does not address mechanisms.**

Response: Thank you for your comment. But, we thought this conclusion could be made according to our results (**e. g., Figures 2, 3; Tables 1, 2**), and we thought it very important for the study.

**Finally, I would still axe the mentions of climate change from the conclusion. The contributions of this paper are on seasonal trends, not climate change. It's not a huge change. It's fine to mention it where it is mentioned just before the conclusion, but it should not be emphasized as the final statement in the paper. The authors justification in the response that because temperature and moisture appear to correlate with the other measured variables, climate change is important is not convincing. Temperature and moisture are always important for microbial processes, and it's a leap to then suggest that this paper provides particular insight on how this relationship will change with climate.**

Response: Thank you for your comment. But, we still thought it was

necessary to mention the climate change in the conclusion. Because, according to this study, significant differences of interannual dynamics of soil microbial biomass and available nitrogen were observed, and that was a comprehensive influence of climate change across 3 years. Both the seasonal trend and interannual pattern of soil microbial biomass and available nitrogen were important in our study. Meanwhile, the alpine ecosystems are sensitive to the future climate change, such as temperature warming and uneven distribution of precipitation. Therefore, it was important to mention the climate change in the conclusion section.

**minor:**

**"An obviously trend of increasing NH4 + –N content was found during the early soil thaw" this revised sentence does not make sense**

Response: Yes, this sentence was revised as "An obviously increased trend of $NH_4^+$–N content was found during the early soil thawing." **(Page 16 line 3)**

Response to Anonymous Referee #2

**I think that thanks to the suggestions of the referees the paper has considerably improved. However I suggest to the authors some further changes, listed below:**

**Pag. 27 line 7: change avilable into available. See also pag 27 line 10.**

Response: Yes, "avilable" was changed into "available" **(Page 4 lines 3)**.

**Pag 28 lines 6-7: Did you add also the soil classification according to the Soil Taxonomy (Silty Loam Inceptisol)? If yes please add the proper reference: Soil Survey Staff. 2014. Keys to Soil Taxonomy, 12th ed. USDA-Natural Resources Conservation Service, Washington, DC.**

Response: Yes, the reference that "Soil Survey Staff: Keys to Soil Taxonomy, 12th ed. USDA-Natural Resources Conservation Service, Washington, DC., 2014." was added **(Page 5 line 17; Page 25 lines 3-4)**.

**Pag 28 lines 10-11: Please specify the elevation of the 3 sites**

Response: Yes, the elevations of the 3 sites were added, i.e., "One site is located at the upper part (3984 m a.s.l.) of the alpine meadow, one at the middle part (3980 m a.s.l.), and one at the lower part (3975m a.s.l.)" **(Page 6 lines 2-3)**.

**Pag 29 line 1: add O horizon before living plant roots and litter**

Response: Yes, "O horizon" was added before "living plant roots and litter" **(Page 6 line 8)**.

**Pag 35 line 14: sorry I don't understand this sentence. What do you mean with "from each other"**

Response: "from each other" means that significant differences compared with each other. We have deleted "from each other" in the revised manuscript.

**Pag 35 line 18: What do you mean woth the term those**

Response: Sorry, we could not find "those" in Page 35 line 18, and we were not sure which "those" in the manuscript did you refer.

**Pag 37 line 10: delete early**

Response: Sorry, we could not find "early" in Page 37 line 10, and we were not sure which "early" in the manuscript did you refer.

**Pag 38 line 18: what do you mean with a-1**

Response: The "$a^{-1}$" in the units for biomass refers to per year, and "$a^{-1}$" was revised as "per year" **(Page 15 lines 3-5)**.

**Pag 40 line 1: Why obviously?**

Response: Sorry, we could not find "obviously" in Page 40 line 1, and we were not sure which "obviously" in the manuscript did you refer.

**Pag 43 line 5: Delete monitoring**

Response: Yes, "monitoring" was deleted **(Page 17 line 14; Page 19 line 2)**.

Response to Anonymous Referee #4

**This paper describes both seasonal and interannual variability in soil microbial biomass and soil available N in alpine tundra with monthly resolution over a 3 year time period. This is an impressive data set which is worthy of publication. I reviewed an earlier version of this manuscript and made a number of suggestions. The writing is much improved but still requires further work. Further, some of my previous comments have not been dealt with to my satisfaction, as described below.**

Response: We thank referee for the helpful comments. After discussing with co-authors, we carefully revised the manuscript and listed in supplement.

**In the previous version I had questions about the MBC/MBN methods as well as the statistics. Neither of these have been dealt with satisfactorily. The methods for determining TDN are still not clear. The procedure for TDN is described on page 7 line 11-16. Line 9-11 describes the chloroform fumigation procedure which is not the methods for TDN but are part of the methods for MBN. TDN is determined on both fumigated and non-fumigated samples (the non-fumigated sample analysis is currently not described in the paper) and the difference is MBN. The chloroform fumigation methods could be moved to the paragraph beginning on page 8 line 6.**

Response: Yes, the methods for determining TDN (**Page 7 lines 5-11**), MBC and MBN (**Page 8 lines 1-13)** were rewrote according to your suggestion

**For the statistics, the response to reviewer 4 indicates that sample ID was included in the model to account for the lack of independence of samples across time. However, the stastistical analysis section of the**

**manuscript does not describe any inclusion of sample ID in the model. Please clarify in the manuscript. Also, all F values throughout the manuscript should include the degrees of freedom.**

Response: Yes. The sample ID was included in the statistical model, i.e., "For the repeat measure analysis with time-dependent covariate, two-way ANOVA was performed via mixed-effects model, with season and year specified as fixed effects, and the sample ID of each sampling time specified as the random effect"(**Page 9 lines 11-13**). The degree of freedom (df) was added behind the *F* value throughout the revised manuscript (**Page 10 line 17; Page 11 lines 9-10; Page 12 lines 5-6, 13-14; Page 13 lines 2-3, 8-9**).

**Some parts of the result section are also not clear. For example, when describing DOC patterns, the seasons are described as being not significantly different from each other (page 11 line 18) and also significantly different (page 12 line 4).**

Response: Thank you for your comment. Although, "the seasonal dynamics of DOC had no significant difference", the interaction effects of DOC between season and year were significantly different. Thus, "the DOC contents during the nongrowing season in 2011–2012 (174.27 mg kg$^{-1}$ ± 32.59 mg kg$^{-1}$) and growing season in 2012–2013 (170.85 mg kg$^{-1}$ ±41.19 mg kg$^{-1}$)" showed the result that "were significantly lower than those in other seasons".

**Also, the results section could be reduced – e.g. The two sentences from page 10 line 17 to page 11 line 1 say the same thing.**

Response: Yes, "the MBC values in the nongrowing seasons were consistently higher than those in the growing seasons" was deleted.

**Finally, just as a suggestion, figure 3 could also be presented with only part A and shading to indicate the different season – this way the data is only presented once rather than repeated in both parts of the figure.**

**The same change could be applied to the other figures. The two types of presentations are presented in the same figure sometimes (Figure 3) and as separate figures in others (Figure 7 and 8).**

Response: Thank you for your comment. But, we thought these figures were indispensable for our study. Because, they showed different information, i.e., the line graphs showed detailed information on intra- and interannual patterns of microbial and nutrient dynamics; the bar graphs showed significant differences in microbial biomass and nutrients between seasons and years, and their interaction effects.

**In the discussion, the authors still need to be cautious about implying causality for some of the patterns they have measured. Some examples are below:**

**Page 13 Line 14 – indicate that the mechanism for the increase in soil MBC and available N are speculative. Also, the conclusion of this paragraph describes an accumulation of organic N which is not described in the remainder of the paragraph.**

Response: Yes, "Thus, an accumulation of inorganic and organic N pools occurred during the long and cold nongrowing seasons in these seasonally frozen ecosystems (Schimel and Mikan, 2005; Schmidt et al., 2007)" was deleted.

**Page 15 Line 15 – Only describe what you have evidence for – e.g. an increase in microbial biomass and not activity**

Response: Yes, the microbial activity was revised as "microbial biomass" **(Page 16 line 1)**.

**And lastly, a few clarifications are required in the discussion:**

**Page 15 line 9 – what do you mean by the "number of bacteria" increased just after thaw? The previous paragraph describes a crash in microbial biomass. Do you mean the proportion of bacteria to fungi? The number of bacterial phenotypes?**

Response: Here, the "number of bacteria" mean the proportion of bacteria, and the "number of bacteria" was revised as "proportion of bacteria" (**Page 15 lines 13-14**).

**Page 15 line 2-5 The units for soil organic matter need an area (per m2?) for this comparison to be relevant. Is this sentence implying that the SOM in tundra is limiting to microbial growth in some circumstances?**

Response: Thank you for your comment. But, we thought the units for soil organic matter did not need an area, because we just need compare the differences of soil organic matter contents among different alpine meadows. Furthermore, we also did not know the values of SOM per square meter ($m^2$) from the references. According to this sentence, we knew that the SOM contents in the alpine meadows of the Qinghai-Tibet Plateau were relatively higher than that in other alpine meadows. It was implied that the available C was relatively sufficient in this region and "might not have restricted microbial activity during the winter–spring transition".

**The writing in this version is much improved over the last version. However, the writing still needs to be improved for clarity and grammar. A number of examples follow:**

**Page 1 Line 10 – replace "occurs seasonally" with "varies seasonally" as the activity occurs all the time**

Response: Yes, "occurs seasonally" was replaced with "varies seasonally" (**Page 1 line 10**).

**Page 1 Line 15 – replace "Topsoil samples were" with "Soil was" and "and were analyzed" with "and analyzed"**

Response: Yes, "Topsoil samples were" and "and were analyzed" were replaced with "Soil was" and "and analyzed", respectively (**Page 1 line 15**).

**Page 1 Line 17 – replace "was measured" with "as measured"**

Response: Yes, "was measured" was replaced with "as measured" (**Page 1 line 16**).

**Page 1 Line 18 – replace "the number of" with "the proportion of"**

Response: Yes, "the number of" was replaced with "the proportion of" (**Page 1 line 18**).

**Page 2 Line 1 – replace "induced by soil temperatures" with "induced by changes in soil temperatures"**

Response: Yes, "induced by soil temperatures" was replaced with "induced by changes in soil temperatures" (**Page 2 line 1**).

**Page 3 Line 11 – delete "apparently"**

Response: Yes, "apparently" was deleted (**Page 3 line 11**).

**Page 4 Line 5 Delete "In alpine systems" as it is repeated later in the sentence**

Response: Yes, "In alpine systems" was deleted (**Page 4 line 5**).

**Page 5 line 8 Delete "and"**

Response: Yes, "and" was deleted (**Page 5 line 8**).

**Page 6 line 10 – Delete "and 15 soil samples… time" as this information is repeated from a few sentences earlier**

Response: Yes, "and 15 soil samples… time" was deleted (**Page 6 line 8**).

**Page 9 line 4 – "analysis of variance" not "variance analysis"**

Response: Yes, "variance analysis" was revised as "analysis of variance" (**Page 9 line 8**).

**Page 14 line 17 – what does the a refer to in the units for biomass? Per year?**

Response: Yes, the "$a^{-1}$" in the units for biomass means "per year", and the "$a^{-1}$" was revised as "per year" (**Page 15 lines 3-5**).

**Similarly, the use of appropriate references is also much improved. A**

**few more which need to be changed are:**

**Page 3, line 3: Mikan is not an alpine reference**

Response: Yes, "Mikan et al., 2002" was deleted and two relevant alpine references were added, i.e., "Brooks et al., 1996; Jefferies et al., 2010" **(Page 3 line 3)**.

**Page 16 line 13 – Qin and Petechey/Gaston are general biodiversity – ecosystem functioning researchers and not appropriate for use in this sentence.**

Response: Yes, "Qin et al., 2003; Petchey and Gaston, 2006" were deleted, and this sentence was deleted **(Page 16 lines 17)**.

Best regards!

Bo Xu

[revised manuscript text omitted]